# Anti-Aliased 2D Gaussian Splatting

**Mae Younes**      **Adnane Boukhayma**
INRIA France, University of Rennes, CNRS, IRISA

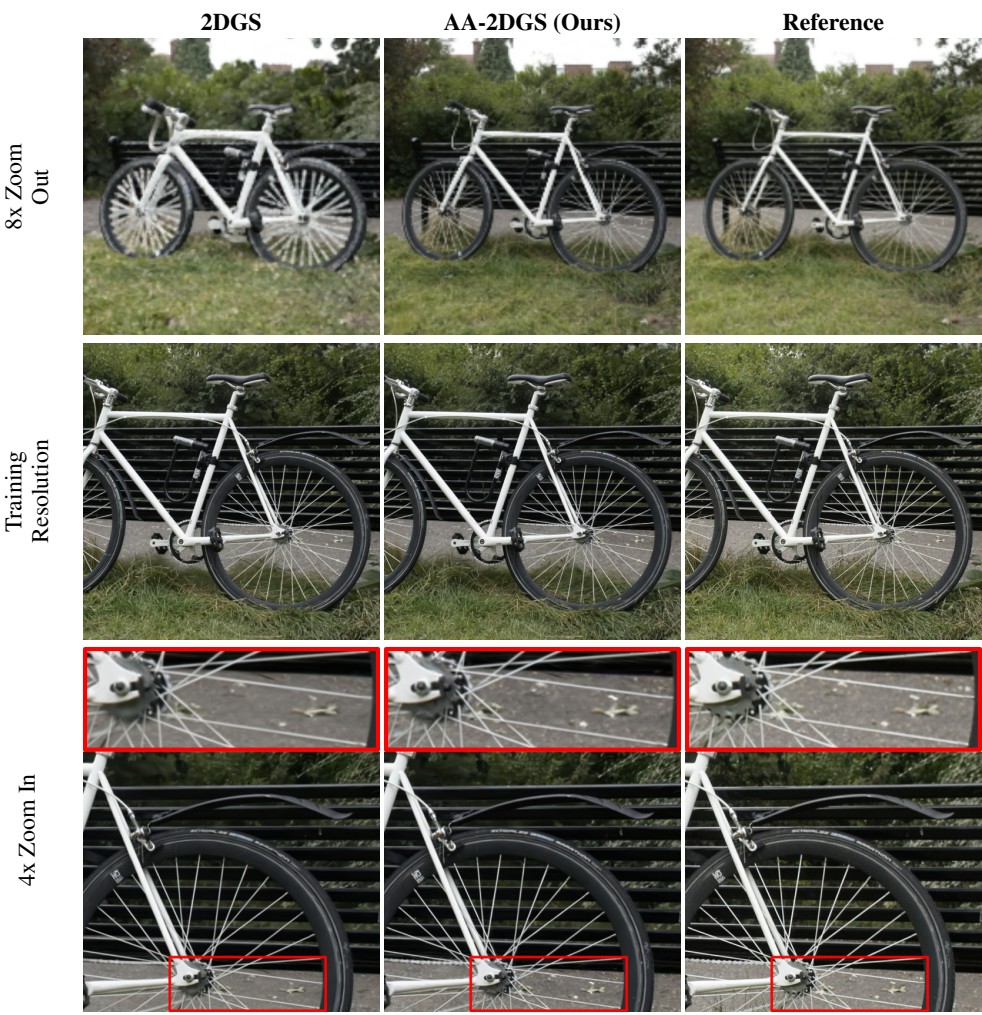

Figure 1: *2DGS and AA-2DGS under change of image sampling rate*. We trained the models on single-scale images and rendered images with different resolutions to simulate Zoom In/Out. While they achieve similar performance at training scale, strong artifacts appear in 2DGS when changing the sampling rate. Our method (AA-2DGS) shows significant improvement in comparison.

## Abstract

2D Gaussian Splatting (2DGS) has recently emerged as a promising method for novel view synthesis and surface reconstruction, offering better view-consistency and geometric accuracy than volumetric 3DGS. However, 2DGS suffers from severe aliasing artifacts when rendering at different sampling rates than those used during training, limiting its practical applications in scenarios requiring camera zoom or varying fields of view. We identify that these artifacts stem from two

39th Conference on Neural Information Processing Systems (NeurIPS 2025).

key limitations: the lack of frequency constraints in the representation and an ineffective screen-space clamping approach. To address these issues, we present AA-2DGS, an anti-aliased formulation of 2D Gaussian Splatting that maintains its geometric benefits while significantly enhancing rendering quality across different scales. Our method introduces a world-space flat smoothing kernel that constrains the frequency content of 2D Gaussian primitives based on the maximal sampling frequency from training views, effectively eliminating high-frequency artifacts when zooming in. Additionally, we derive a novel object-space Mip filter by leveraging an affine approximation of the ray-splat intersection mapping, which allows us to efficiently apply proper anti-aliasing directly in the local space of each splat. Code will be available at `AA-2DGS`.

## 1 Introduction

3D reconstruction from multi-view images has been a fundamental problem in computer vision, graphics, and machine learning for decades. This field has seen renewed interest due to its applications in autonomous driving, medical imaging, gaming, visual effects, and extended reality experiences, which all require high-quality 3D modeling and visualization.

Neural Radiance Fields (NeRF) [43] revolutionized this area by introducing a neural representation that models scenes through differentiable volume rendering. Building on this foundation, 3D Gaussian Splatting (3DGS) [31] recently revitalized point-based graphics by replacing neural networks with explicit 3D Gaussian primitives. These primitives are rasterized and rendered via volume resampling and their parameters are optimized through gradient-based inverse rendering. With its efficient density control, primitive sorting, and tile-based rasterization, 3DGS achieves state-of-the-art novel view synthesis while enabling real-time rendering and requiring shorter training times.

The Gaussian Splatting approach has evolved into two primary variants: 3DGS [31] and 2D Gaussian Splatting (2DGS) [23]. While 3DGS represents scenes using volumetric 3D Gaussian primitives, 2DGS employs flattened 2D Gaussian disks embedded in 3D space. This distinction is significant: 3DGS projects 3D Gaussians onto the screen to obtain 2D screen-space Gaussians, which are then rendered. In contrast, 2DGS evaluates the Gaussians directly at ray-splat intersections in the local coordinates of each planar primitive. This approach gives 2DGS superior geometric accuracy, particularly for depth and normal reconstruction, making it valuable for applications requiring precise geometry such as mesh recovery [10], physics-based rendering [18], and reflectance modeling [69].

Despite its strengths, 2DGS faces a significant challenge: its formulation complicates the integration of proper anti-aliasing techniques. The 2DGS method attempts to address this by employing a screen-space lower bounding approach (clamping) [5], but our investigation reveals that this approach often exacerbates aliasing artifacts rather than mitigating them. This is particularly evident when rendering at different sampling rates, such as zooming in or out from a scene (See Tab. 1, Tab. 2 and 3).

Recent work on Mip-Splatting [75] has identified two key sources of aliasing in 3DGS: the lack of 3D frequency constraints and inadequate screen-space filtering. Mip-Splatting addresses these issues by introducing a 3D smoothing filter to constrain the frequency content of primitives based on training view sampling rates, and by replacing the traditional screen-space dilation with a Mip filter that better approximates the physical imaging process. However, these solutions cannot be directly applied to 2DGS due to its fundamentally different primitive representation and rendering approach.

In this paper, we present Anti-Aliased 2D Gaussian Splatting (AA-2DGS), an approach that makes two key contributions: • We introduce a world-space flat smoothing kernel that constrains the frequency content of 2D Gaussian primitives based on the sampling rates of the training views. This addresses high-frequency artifacts when zooming in on a scene by ensuring that the primitives respect the Nyquist-Shannon sampling theorem [55]. • We derive an object-space Mip filter that leverages an affine approximation of the ray-splat intersection mapping used in 2DGS. This allows us to incorporate Mip filtering directly in the local space of each splat, where the Gaussian evaluation occurs. The resulting formulation is both mathematically elegant and computationally efficient. Ablative analysis of these two components is in the supplementary material.

We evaluate AA-2DGS on standard novel view synthesis datasets, including Mip-NeRF 360 [2] and Blender [43], as well as the DTU [29] mesh reconstruction benchmark. Our results demonstrate that AA-2DGS consistently outperforms the original 2DGS method, particularly under challenging

conditions such as varying sampling rates and mixed resolution training. Importantly, our approach maintains the geometric accuracy that makes 2DGS valuable while significantly reducing aliasing artifacts.

## 2 Related Work

Until recently, implicit representations coupled with differentiable volume rendering have been at the forefront of 3D shape and appearance modeling. NeRFs [43] model scenes with an implicit density and view dependent radiance, parametrized with MLPs. Anti-aliasing can be implemented in these representations through cone tracing and pre-filtering the input positional or feature encodings [1, 2, 3, 22, 81]. Multiscale volume rendering requires intensive MLP querying, thus limiting the rendering frame rate. This issue can be alleviated with grid based representations [44, 58, 14, 15, 8]. These can struggle with large unbounded reconstruction, despite Level-of-detail Octrees [40]. By expressing density as a function of a signed distance field, NeRFs lead to powerful geometry recovery methods [62, 70, 38, 73, 26, 66]. Implicit reconstruction has been robustified against noise and sparsity from both image and point cloud input using generalizable data priors (e.g. [74, 9, 30, 36, 46, 24, 48, 49, 52]) and various regularizations (e.g. [45, 68, 25, 12, 37, 51, 47, 50, 49, 4, 17]).

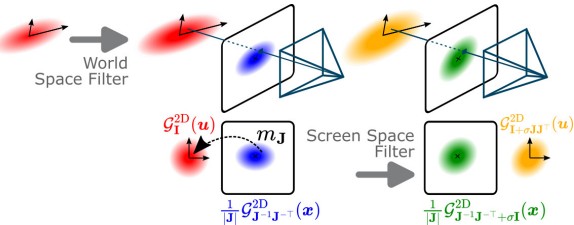

Figure 2: *Overview.* We constrain the maximum frequency of our 2D Gaussians (Red) to a limit estimated from the training images with a world-space flat smoothing filter. Next, leveraging an affine approximation of the mapping from screen space to local splat space: $m_{\mathbf{J}}$ where $\mathbf{J} = \frac{\partial \mathbf{u}}{\partial \mathbf{x}}$, we can express the reconstruction kernel footprint in screen space (Blue). This enables the integration of a screen space anti-aliasing Gaussian filter (Green). Via the affine mapping, we can revert to a final simpler and computationally lighter expression of our kernel (Orange) defined in local splat space.

3D Gaussian splatting [31] subverted this trend lately. Combining volume rendering [42] and EWA splatting [82] within an efficient inverse rendering optimization [33]. It has spawned substantial research due to its remarkable novel view performance and high rendering frame rate. Extensions include generalizable models [7, 41, 28, 60], bundle adjustment [80, 16, 27, 71], higher dimensional primitives [13], more expressive texture splatting [54, 59, 6, 72], spatiotemporal models [65], in addition to several methods to improve density control [32, 61, 78], model compactness [35, 63] and training speed [34, 79, 20]. Recent work augmented 3DGS's anti-aliasing abilities. [39] analytically approximates the integral of Gaussian signals over pixel areas using a conditioned logistic function. However, calculating integrals for every pixel can be computationally intensive, especially for high-resolution images and large-scale scenes. [67] represents the scene with Gaussians at multiple scales, rendering higher-resolution images with smaller Gaussians and lower-resolution images with fewer larger ones. This strategy can lead to important memory overheads nonetheless. [57] uses a frustum-based supersampling strategy to mitigate aliasing, which can be computationally costly, especially at higher resolutions. Closest to our contribution, [75] reinstated the EWA screen space filter in 3DGS, and proposed to use a 3D low-pass filter to band-limit the 3D Gaussian representation based on the sampling limits of the input images. The 2DGS [23] representation uses planar primitives instead of volumetric ones. It offers competitive novel view synthesis and state-of-the-art mesh reconstruction performance, where 3DGS fails to faithfully recover depth. To the best of our knowledge, ours is the first work that analyses the anti-aliasing capabilities of 2DGS, and proposes a solution to its limitations in this department.

## 3 Method

Our method extends the 2DGS framework by incorporating frequency-based filtering techniques to address aliasing artifacts across varying sampling rates. We first review the sampling theorem and 2D Gaussian Splatting to establish the foundation for our antialiasing techniques. Then, we introduce our key contributions: (1) a world-space flat smoothing kernel that effectively limits the frequency of the 2D Gaussian primitives based on the sampling rate of training views, and (2) an object-space Mip filter that leverages the ray-splat intersection mapping to accurately perform antialiasing directly in the local space of each splat.

### 3.1 Preliminaries

**Sampling Theorem**  The Nyquist-Shannon Sampling Theorem [55] states that a band-limited signal with no frequency components above $\nu$ can be perfectly reconstructed from samples taken at a rate $\hat{\nu} \geq 2\nu$. Otherwise, aliasing occurs as high frequencies are incorrectly mapped to lower ones. To prevent this, a low-pass filter is applied prior to sampling to suppress frequencies above $\frac{\hat{\nu}}{2}$. This principle guides our antialiasing strategy in 2D Gaussian Splatting.

**2D Gaussian Splatting**  2DGS [23] represents scenes with oriented planar Gaussian disks in 3D space. Each 2DGS primitive has a center $\mathbf{p}_k \in \mathbb{R}^3$, two orthogonal tangential vectors $\mathbf{t}_u$ and $\mathbf{t}_v$ and scaling factors $\mathbf{s} = (s_u, s_v)$. Using rotation matrix $\mathbf{R} = [\mathbf{t}_u, \mathbf{t}_v, \mathbf{t}_u \times \mathbf{t}_v]$ and scaling matrix $\mathbf{S}$ the geometry of the primitive is defined in the local tangent plane parameterized by:

$$P(u,v) = \mathbf{p}_k + s_u \mathbf{t}_u u + s_v \mathbf{t}_v v = \mathbf{H}(u,v,1,1)^\top, \tag{1}$$

$$\text{where} \quad \mathbf{H} = \begin{bmatrix} s_u \mathbf{t}_u & s_v \mathbf{t}_v & \mathbf{0} & \mathbf{p}_k \\ 0 & 0 & 0 & 1 \end{bmatrix} = \begin{bmatrix} \mathbf{RS} & \mathbf{p}_k \\ \mathbf{0} & 1 \end{bmatrix}. \tag{2}$$

For a point $\mathbf{u} = (u,v)$ in the primitive local space, the kernel value writes:

$$\mathcal{G}_{\mathbf{I}}^{2D}(\mathbf{u}) = \exp\left(-\frac{u^2 + v^2}{2}\right), \tag{3}$$

where $\mathbf{I}$ is the identity matrix, representing the covariance of the Gaussian in its local coordinate system. Rendering is preformed via volumetric alpha blending using primitive opacity $\alpha_i$ and color $\mathbf{c}_i$:

$$\mathbf{C}(\mathbf{x}) = \sum_{i=1} \mathbf{c}_i \, \alpha_i \, \mathcal{G}_i(\mathbf{u}(\mathbf{x})) \prod_{j=1}^{i-1} (1 - \alpha_j \, \mathcal{G}_j(\mathbf{u}(\mathbf{x}))). \tag{4}$$

**Ray-Splat Intersection**  2DGS employs a ray-splat intersection method based on [56, 64] for rendering. Given an image coordinate $\mathbf{x} = (x,y)$, the splat intersection is the intersection of the $x$-plane, $y$-plane, and the splat plane. Homogeneous representations of the $x$-plane and $y$-plane are $\mathbf{h}_x = (-1, 0, 0, x)^\top$ and $\mathbf{h}_y = (0, -1, 0, y)^\top$. These planes are transformed into the local coordinate system of the splat using:

$$\mathbf{h}_u = (\mathbf{WH})^\top \mathbf{h}_x, \quad \mathbf{h}_v = (\mathbf{WH})^\top \mathbf{h}_y, \tag{5}$$

where $\mathbf{W}$ is the world to screen space transform matrix. The intersection point in local coordinates $\mathbf{u}(\mathbf{x})$ then writes:

$$u(\mathbf{x}) = \frac{\mathbf{h}_u^2 \mathbf{h}_v^4 - \mathbf{h}_u^4 \mathbf{h}_v^2}{\mathbf{h}_u^1 \mathbf{h}_v^2 - \mathbf{h}_u^2 \mathbf{h}_v^1}, \quad v(\mathbf{x}) = \frac{\mathbf{h}_u^4 \mathbf{h}_v^1 - \mathbf{h}_u^1 \mathbf{h}_v^4}{\mathbf{h}_u^1 \mathbf{h}_v^2 - \mathbf{h}_u^2 \mathbf{h}_v^1}, \tag{6}$$

where $\mathbf{h}_u^i$ and $\mathbf{h}_v^i$ denote the $i$-th component of the 4D planes.

**Antialiasing Challenges in 2DGS**  The original 2DGS implementation addresses the issue of degenerate cases (when a Gaussian is viewed from a slanted angle) by employing an object-space low-pass filter:

$$\hat{\mathcal{G}}(\mathbf{x}) = \max\left\{\mathcal{G}_{\mathbf{I}}^{2D}(\mathbf{u}(\mathbf{x})), \mathcal{G}_{\mathbf{I}}^{2D}(\frac{\mathbf{x} - \mathbf{c}}{\sigma})\right\} \tag{7}$$

where $\mathbf{c}$ is the projection of the center $\mathbf{p}_k$ and $\sigma$ is a scaling factor. This mechanism was inspired by the heuristic EWA approximation in [5] that was proposed to handle minification and aliasing when EWA filtering is not possible.

While clamping improves rendering stability, it has notable drawbacks. First, the use of a $\max$ operation introduces discontinuities in the gradient flow, potentially hindering optimization. Second, the conditional logic in Eq. 7 can cause thread divergence in CUDA warps, reducing GPU efficiency. Third, the heuristic compares distances at different domains (local splat space vs. screen space) and lacks the antialiasing quality of true screen space EWA filtering. Even standard EWA can suffer from aliasing and over blurriness (as demonstrated by Mip-Splatting [75]), issues worsened by this approximation. In the following, we present our solution to these challenges.

## 3.2 World-Space Flat Smoothing Kernel

The goal here is to constrain the maximum frequency of the 3D representation during optimization based on the Nyquist limit of training views, as highlighted by [75]. However, unlike 3DGS, our primitives are planar. Therefore, we need to adapt the 3D smoothing filter concept to our flattened primitive representation.

**Multiview Frequency Bounds**   Following the analysis in [75], we determine the maximal sampling rate for each primitive based on the training views. For an image with focal length $f$ in pixel units, the world-space sampling interval $\hat{T}$ at depth $d$ is $\hat{T} = \frac{1}{\hat{\nu}} = \frac{d}{f}$ where $\hat{\nu}$ is the sampling frequency. We determine the maximal sampling rate for primitive $k$ as:

$$\hat{\nu}_k = \max_{n=1...N} \left\{ \mathbb{1}_n(\mathbf{p}_k) \cdot \frac{f_n}{d_n} \right\}, \tag{8}$$

where $N$ is the total number of training images, and $\mathbb{1}_n(\mathbf{p})$ is an indicator function that evaluates to true if the Gaussian center $\mathbf{p}_k$ falls within the view frustum of the $n$-th camera.

**Flat Smoothing**   The 3D smoothing filter in [75] convolves each 3D primitive Gaussian $\mathcal{G}_{\boldsymbol{\Sigma}_k}$ with an isotropic 3D low-pass filter $\mathcal{G}_{\text{low}} = \mathcal{G}_{\sigma^2_{\text{smooth},k}\mathbf{I}_3}$, with $\sigma^2_{\text{smooth},k} = \frac{s_{\text{reg}}}{\hat{\nu}_k^2}$, $s_{\text{reg}}$ being a hyperparameter. This results in a 3D Gaussian with covariance $\boldsymbol{\Sigma}_k + \sigma^2_{\text{smooth},k}\mathbf{I}_3$.

Our 2D Gaussian primitives are embedded on 2D planes. In the splat plane coordinate system (spanned by $\mathbf{t}_{u_k}, \mathbf{t}_{v_k}$ centered at $\mathbf{p}_k$), this primitive intrinsically represents a 2D Gaussian distribution with covariance $\mathbf{V}_k = \text{diag}(s^2_{u_k}, s^2_{v_k})$. To achieve a similar band-limiting effect while keeping the primitive flat, we project the isotropic 3D smoothing kernel $\mathcal{G}_{\text{low}}$ onto the plane of the 2D Gaussian primitive. This projection yields an isotropic 2D Gaussian filter with the same variance $\sigma^2_{\text{smooth},k}\mathbf{I}_2$ in the planar coordinates defined by $(\mathbf{t}_{u_k}, \mathbf{t}_{v_k})$. Next we convolve the primitive's intrinsic 2D Gaussian (covariance $\mathbf{V}_k$) with this projected 2D smoothing filter (covariance $\sigma^2_{\text{smooth},k}\mathbf{I}_2$), both on the splat's plane. This convolution yields a new 2D Gaussian on the same plane with covariance:

$$\mathbf{V}_k^{\text{eff}} = \mathbf{V}_k + \sigma^2_{\text{smooth},k}\mathbf{I}_2 = \begin{pmatrix} s^2_{u_k} + \sigma^2_{\text{smooth},k} & 0 \\ 0 & s^2_{v_k} + \sigma^2_{\text{smooth},k} \end{pmatrix}. \tag{9}$$

To maintain energy conservation, the primitive's opacity $\alpha_k$ is modulated, analogously to [75]. For unnormalized Gaussians, this factor is the ratio of the product of scales:

$$\alpha_k^{\text{smooth}} = \alpha_k \frac{s_{u_k} s_{v_k}}{\sqrt{s^2_{u_k} + \sigma^2_{\text{smooth},k}} \cdot \sqrt{s^2_{v_k} + \sigma^2_{\text{smooth},k}}}. \tag{10}$$

The maximal sampling rates $\hat{\nu}_k$, and thus $\sigma^2_{\text{smooth},k}$, are computed based on the training views and remain fixed during testing. This world-space flat smoothing effectively regularizes the 2D primitives by ensuring their footprint on their respective planes adheres to the sampling limits, preventing high-frequency artifacts when zooming in, analogous to its 3D counterpart.

## 3.3 Object-Space Mip Filter

While the flat smoothing kernel addresses pre-aliasing from the representation itself, we also need to handle aliasing during rendering, especially when projecting splats to screen resolutions that differ from training (e.g., zooming out). Standard 3DGS and Mip-Splatting apply screen space filters. However, 2DGS evaluates Gaussians at ray-splat intersection points $\mathbf{u}_k(\mathbf{x})$ in the splat's local space, making direct application of a screen space filter non-trivial.

**Ray-Splat Intersection Affine Mapping**   The key insight of our approach is to leverage the ray-splat intersection mapping used in 2DGS and derive an affine approximation of it, adapting the principles of Elliptical Weighted Average (EWA) filtering [21, 82, 53] to the 2DGS framework. This allows us to map a screen space Mip filter to the local space of each splat, where the Gaussian evaluation actually happens.

Let $m$ be the mapping from pixel coordinates $\mathbf{x}$ to local splat coordinates $\mathbf{u}$. Let us approximate this mapping using a first-order Taylor expansion around a pixel location $\mathbf{x}_0$:

$$m(\mathbf{x}) \approx m(\mathbf{x}_0) + \mathbf{J} \cdot (\mathbf{x} - \mathbf{x}_0) = \mathbf{u}_0 + \mathbf{J}(\mathbf{x} - \mathbf{x}_0), \tag{11}$$

where $\mathbf{u}_0 = m(\mathbf{x}_0)$ is the intersection of the ray passing through $\mathbf{x}_0$ with the splat, and $\mathbf{J}$ is the Jacobian of the mapping evaluated at $\mathbf{x}_0$. It captures how the local coordinates change with respect to small changes in pixel coordinates, and can be computed analytically from the derivation of the ray-splat intersection formula (Eq.6):

$$\mathbf{J} = \begin{bmatrix} \frac{\partial u}{\partial x} & \frac{\partial u}{\partial y} \\ \frac{\partial v}{\partial x} & \frac{\partial v}{\partial y} \end{bmatrix}. \tag{12}$$

**Mip Filter Mapping**  Using properties of Gaussian functions under affine transformations, we can express the 2D Gaussian in screen space as:

$$\mathcal{G}_{\mathbf{I}}^{2D}(\mathbf{u}) = \mathcal{G}_{\mathbf{I}}^{2D}(m(\mathbf{x})) = \frac{1}{|\mathbf{J}|}\mathcal{G}_{\mathbf{J}^{-1}\mathbf{J}^{-\top}}^{2D}(\mathbf{x}). \tag{13}$$

To perform antialiasing, we convolve this transformed Gaussian with a Mip filter in screen space. Similar to Mip-Splatting, we use a Gaussian Mip filter with covariance $\sigma\mathbf{I}$ to approximate the box filter of the physical imaging process, but we note that we can also use the EWA filter here:

$$\mathcal{G}_{\text{mip}}^{2D}(\mathbf{x}) = \left(\frac{1}{|\mathbf{J}|}\mathcal{G}_{\mathbf{J}^{-1}\mathbf{J}^{-\top}}^{2D} \otimes \mathcal{G}_{\sigma\mathbf{I}}^{2D}\right)(\mathbf{x}). \tag{14}$$

Using the property that the convolution of two Gaussians results in another Gaussian with the sum of their covariance matrices, we get:

$$\mathcal{G}_{\text{mip}}^{2D}(\mathbf{x}) = \frac{1}{|\mathbf{J}|}\mathcal{G}_{\mathbf{J}^{-1}\mathbf{J}^{-\top}+\sigma\mathbf{I}}^{2D}(\mathbf{x}). \tag{15}$$

**Mapping Back to Object Space**  While we could evaluate the Mip filtered Gaussian directly in screen space, it is more efficient to map it back to the local space of the splat. Using the properties of Gaussian functions under affine transformations again, we get:

$$\mathcal{G}_{\text{mip}}^{2D}(\mathbf{x}) = \mathcal{G}_{\mathbf{I}+\sigma\mathbf{J}\mathbf{J}^{\top}}^{2D}(\mathbf{u}). \tag{16}$$

We denote the new covariance in local space: $\mathbf{\Sigma}'_{\text{local},k}(\mathbf{x}) = \mathbf{I} + \sigma\mathbf{J}\mathbf{J}^{\top}$. The mip filtered Gaussian contribution for splat $k$ at pixel $\mathbf{x}$ is then evaluated in local $uv$-space at $\mathbf{u}_k(\mathbf{x})$:

$$\mathcal{G}_{\text{obj-mip},k}(\mathbf{x}) = \sqrt{\frac{|\mathbf{I}_2|}{|\mathbf{\Sigma}'_{\text{local},k}(\mathbf{x})|}} \exp\left(-\frac{1}{2}\mathbf{u}_k(\mathbf{x})^{\top}(\mathbf{\Sigma}'_{\text{local},k}(\mathbf{x}))^{-1}\mathbf{u}_k(\mathbf{x})\right). \tag{17}$$

Our object-space Mip filter eliminates the computational overhead and numerical instabilities of screen space evaluation. Unlike object-space EWA splatting [53], which approximates perspective around the primitive center, we center the affine approximation per pixel for improved accuracy, especially with large primitives or challenging views.

## 4  Experiments

We evaluate our work through novel view synthesis on datasets Blender [43] and Mip-NeRF 360 [2] following the benchmark in Mip-Splatting [75]. These experiments assess generalization to both in and out of distribution pixel sampling rate. We additionally evaluate our work through the 3D surface reconstruction experiment on dataset DTU [29] following the benchmark in [23]. We provide **additional results, ablations and an extended discussion on limitations in the supplementary material**.

### 4.1  Implementation Details

We build our method upon the open-source implementation of 2DGS. Following Mip-Splatting, we train our models for 30K iterations across all scenes and use the same loss function, Gaussian density control strategy, schedule, and hyperparameters. For novel view synthesis experiments, we disable the depth and normal regularizations used by 2DGS and enable them for surface reconstruction experiment. We follow the Mip-Splatting approach and recompute the sampling rate of each 2D Gaussian primitive every $m = 100$ iterations. Similarly, we choose the variance of our object-space Mip filter as $0.1$, approximating a single pixel, and the variance of the flat smoothing filter as $0.2$. We implement our object-space Mip filtering with custom CUDA kernels for both forward and backward computation. Due to the extra computations required by the Mip filter, our approach incurs an overhead of 15-30% in rendering time compared to the aliased 2DGS. We conduct all experiments on NVIDIA RTX A6000 GPUs.

## 4.2 Evaluation on the Blender Dataset

The Blender dataset [43] includes 8 synthetically rendered scenes with complex geometry and realistic materials. Each scene has 100 training views and 200 test views, rendered at 800×800 resolution.

**Multi-scale Training and Multi-scale Testing**  We train our model with multi-scale data and evaluate with multi-scale data following previous work [1, 22, 75]. We adopt the biased sampling strategy in [75, 1, 22] where rays from full-resolution images are sampled at a higher frequency (40%) compared to those from lower resolutions (20% per remaining resolution level). This ensures greater emphasis on high-resolution data while maintaining coverage across all image scales. Table 1 shows the quantitative results of this experiment. Except for 2DGS variants, we report numbers for other methods from [75]. We outperform the 3DGS based Mip-Splatting and state-of-the-art Nerf based methods MipNeRF and Tri-MipRF on average PSNR, and also almost across most scales. Notice that we outperform the 2DGS baselines with a large margin across all scales. This shows that our Object-Space Mip filter enables the model to handle varying levels of detail without overfitting on a single scale. On the other hand, we showcase the finding that the screen space clamping heuristic hinders the performance of vanilla 2DGS at the higher scales.

| | PSNR ↑ | | | | | SSIM ↑ | | | | | LPIPS ↓ | | | | |
|---|---|---|---|---|---|---|---|---|---|---|---|---|---|---|---|
| | Full Res. | ¹/₂ Res. | ¹/₄ Res. | ¹/₈ Res. | Avg. | Full Res. | ¹/₂ Res. | ¹/₄ Res. | ¹/₈ Res. | Avg. | Full Res. | ¹/₂ Res. | ¹/₄ Res. | ¹/₈ Res | Avg. |
| NeRF w/o $\mathcal{L}_{area}$ [43, 1] | 31.20 | 30.65 | 26.25 | 22.53 | 27.66 | 0.950 | 0.956 | 0.930 | 0.871 | 0.927 | 0.055 | 0.034 | 0.043 | 0.075 | 0.052 |
| NeRF [43] | 29.90 | 32.13 | 33.40 | 29.47 | 31.23 | 0.938 | 0.959 | 0.973 | 0.962 | 0.958 | 0.074 | 0.040 | 0.024 | 0.039 | 0.044 |
| MipNeRF [1] | 32.63 | 34.34 | 35.47 | 35.60 | 34.51 | 0.958 | 0.970 | 0.979 | 0.983 | 0.973 | 0.047 | 0.026 | 0.017 | 0.012 | 0.026 |
| Plenoxels [15] | 31.60 | 32.85 | 30.26 | 26.63 | 30.34 | 0.956 | 0.967 | 0.961 | 0.936 | 0.955 | 0.052 | 0.032 | 0.045 | 0.077 | 0.051 |
| TensoRF [8] | 32.11 | 33.03 | 30.45 | 26.80 | 30.60 | 0.956 | 0.966 | 0.962 | 0.939 | 0.956 | 0.056 | 0.038 | 0.047 | 0.076 | 0.054 |
| Instant-NGP [44] | 30.00 | 32.15 | 33.31 | 29.35 | 31.20 | 0.939 | 0.961 | 0.974 | 0.963 | 0.959 | 0.079 | 0.043 | 0.026 | 0.040 | 0.047 |
| Tri-MipRF [22] | 32.65 | 34.24 | 35.02 | 35.53 | 34.36 | 0.958 | 0.971 | 0.980 | 0.987 | 0.974 | 0.047 | 0.027 | 0.018 | 0.012 | 0.026 |
| 3DGS [31] | 28.79 | 30.66 | 31.64 | 27.98 | 29.77 | 0.943 | 0.962 | 0.972 | 0.960 | 0.960 | 0.065 | 0.038 | 0.025 | 0.031 | 0.040 |
| 3DGS [31] + EWA [82] | 31.54 | 33.26 | 33.78 | 33.48 | 33.01 | 0.961 | 0.973 | 0.979 | 0.983 | 0.974 | 0.043 | 0.026 | 0.021 | 0.019 | 0.027 |
| Mip-Splatting [75] | 32.81 | 34.49 | 35.45 | 35.50 | 34.56 | 0.967 | 0.977 | 0.983 | 0.988 | 0.979 | 0.035 | 0.019 | 0.013 | 0.010 | 0.019 |
| 2DGS [23] | 28.58 | 30.24 | 31.42 | 27.35 | 29.40 | 0.938 | 0.959 | 0.970 | 0.952 | 0.954 | 0.078 | 0.047 | 0.029 | 0.042 | 0.049 |
| 2DGS w/o Clamping | 31.64 | 33.33 | 31.61 | 27.62 | 31.05 | 0.960 | 0.973 | 0.973 | 0.957 | 0.966 | 0.043 | 0.023 | 0.028 | 0.056 | 0.038 |
| AA-2DGS (ours) | 32.68 | 34.53 | 35.65 | 35.53 | 34.60 | 0.965 | 0.976 | 0.983 | 0.988 | 0.978 | 0.037 | 0.02 | 0.013 | 0.010 | 0.020 |

Table 1: **Multi-scale Training and Multi-scale Testing on the Blender dataset [43].** Our approach significantly improves 2DGS and achieves comparable or better performance than Mip-Splatting.

**Single-scale Training and Multi-scale Testing**  Following [75], we train on full resolution images and test at various lower resolutions (1×, 1/2, 1/4, 1/8) to mimic zoom-out effects. Table 2 shows the quantitative results of this experiment. The Clamping deteriorates the performance of 2DGS in this experiment as well. Our method outperforms all anti-aliased 3DGS and NeRF based competition in average PSNR and also across all resolutions, with a large improvement with respect to the baseline 2DGS. This is a testimony of the effectiveness of our Object-Space Mip filter combined with the accurate ray splat intersection rendering. These results are clearly reflected in the qualitative superiority of our renderings especially at lower resolutions compared to training, as shown in Figure 3, or also the zooming out visualization in the teaser Figure 1.

| | PSNR ↑ | | | | | SSIM ↑ | | | | | LPIPS ↓ | | | | |
|---|---|---|---|---|---|---|---|---|---|---|---|---|---|---|---|
| | Full Res. | ¹/₂ Res. | ¹/₄ Res. | ¹/₈ Res. | Avg. | Full Res. | ¹/₂ Res. | ¹/₄ Res. | ¹/₈ Res. | Avg. | Full Res. | ¹/₂ Res. | ¹/₄ Res. | ¹/₈ Res | Avg. |
| NeRF [43] | 31.48 | 32.43 | 30.29 | 26.70 | 30.23 | 0.949 | 0.962 | 0.964 | 0.951 | 0.956 | 0.061 | 0.041 | 0.044 | 0.067 | 0.053 |
| MipNeRF [1] | 33.08 | 33.31 | 30.91 | 27.97 | 31.31 | 0.961 | 0.970 | 0.969 | 0.961 | 0.965 | 0.045 | 0.031 | 0.036 | 0.052 | 0.041 |
| TensoRF [8] | 32.53 | 32.91 | 30.01 | 26.45 | 30.48 | 0.960 | 0.969 | 0.965 | 0.948 | 0.961 | 0.044 | 0.031 | 0.044 | 0.073 | 0.048 |
| Instant-NGP [44] | 33.09 | 33.00 | 29.84 | 26.33 | 30.57 | 0.962 | 0.969 | 0.964 | 0.947 | 0.961 | 0.044 | 0.033 | 0.046 | 0.075 | 0.049 |
| Tri-MipRF [22] | 32.89 | 32.84 | 28.29 | 23.87 | 29.47 | 0.958 | 0.967 | 0.951 | 0.913 | 0.947 | 0.046 | 0.033 | 0.046 | 0.075 | 0.050 |
| 3DGS [31] | 33.33 | 26.95 | 21.38 | 17.69 | 24.84 | 0.969 | 0.949 | 0.875 | 0.766 | 0.890 | 0.030 | 0.032 | 0.066 | 0.121 | 0.063 |
| 3DGS [31] + EWA [82] | 33.51 | 31.66 | 27.82 | 24.63 | 29.40 | 0.969 | 0.971 | 0.959 | 0.940 | 0.960 | 0.032 | 0.024 | 0.033 | 0.047 | 0.034 |
| Mip-Splatting [75] | 33.36 | 34.00 | 31.85 | 28.67 | 31.97 | 0.969 | 0.977 | 0.978 | 0.973 | 0.974 | 0.031 | 0.019 | 0.019 | 0.026 | 0.024 |
| 2DGS [23] | 33.05 | 27.64 | 20.61 | 16.59 | 24.47 | 0.969 | 0.952 | 0.856 | 0.720 | 0.874 | 0.033 | 0.037 | 0.082 | 0.151 | 0.076 |
| 2DGS w/o Clamping | 33.18 | 33.04 | 29.74 | 26.21 | 30.54 | 0.968 | 0.973 | 0.964 | 0.945 | 0.963 | 0.032 | 0.024 | 0.040 | 0.054 | 0.042 |
| AA-2DGS (ours) | 33.24 | 34.10 | 32.11 | 29 | 32.11 | 0.967 | 0.976 | 0.978 | 0.973 | 0.974 | 0.034 | 0.020 | 0.019 | 0.024 | 0.024 |

Table 2: **Single-scale Training and Multi-scale Testing on the Blender Dataset [43].** All methods are trained on full-resolution images and evaluated at four different (smaller) resolutions, with lower resolutions simulating zoom-out effects. AA-2DGS yields comparable results at training resolution to 3DGS-based methods and achieves significant improvements compared to other methods in almost all metrics at different lower scales.

## 4.3 Evaluation on the Mip-NeRF 360 Dataset

The Mip-NeRF 360 Dataset [2] is designed to evaluate rendering methods in unbounded, real-world 360° scenes with complex backgrounds, varying lighting, and challenging view-dependent effects. It consists of 9 real-world indoor and outdoor scenes. Each scene contains 100 to 400 training images and 200 test images.

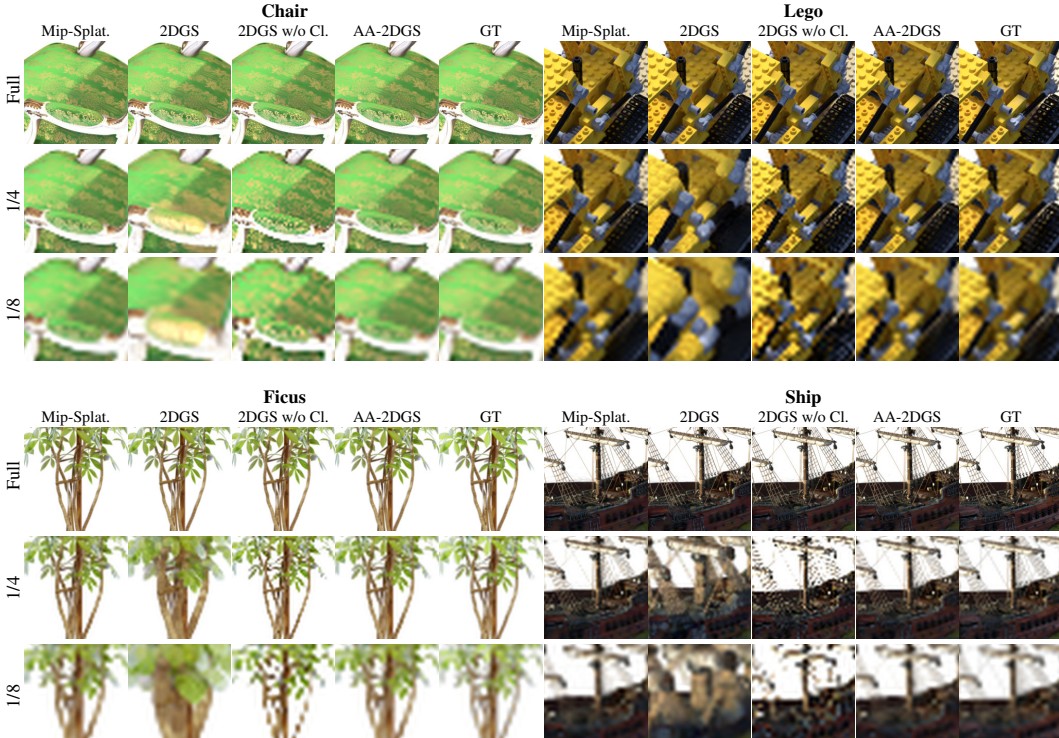

Figure 3: **Single-scale Training and Multi-scale Testing on the Blender Dataset [43].** All methods are trained at full resolution and evaluated at different (smaller) resolutions to mimic zoom-out. Our method (AA-2DGS) consistently demonstrates improved quality across all sampling rates compared to the baseline 2DGS method.

**Single-scale Training and Multi-scale Testing**  Following [75], we train here on 1/8 resolution images and test at various higher resolutions ($1\times$, $2\times$, $4\times$, $8\times$) to simulate zoom-in effects. Results are shown in Table 3. We perform on par here with the state-of-the-art anti-aliased 3DGS, with considerable improvement as compared to the NeRF based methods due to their MLP overfitting. Removing the clamping from 2DGS increases its performance. Our healthy margins with respect to the 2DGS baselines demonstrate the utility of our flat smoothing kernel for frequency regularization. This can be visualized in the qualitative comparison of Figure 4, where AA-2DGS shows reduced aliasing artifacts compared to its baselines and renders fine details with more fidelity without aliasing. We also find that the clamping heuristic hurts the performance of vanilla 2DGS. We note that while the flat smoothing kernel improves results for this magnification experiment, the nature of 2D planar primitives makes them more likely to become extremely thin during training at low resolution. When rendered at higher resolutions, they appear as "needle-like" artifacts because they're too small/thin relative to the display resolution.

| | PSNR ↑ | | | | | SSIM ↑ | | | | | LPIPS ↓ | | | | |
|---|---|---|---|---|---|---|---|---|---|---|---|---|---|---|---|
| | 1× Res. | 2× Res. | 4× Res. | 8× Res. | Avg. | 1× Res. | 2× Res. | 4× Res. | 8× Res. | Avg. | 1× Res. | 2× Res. | 4× Res. | 8× Res. | Avg. |
| Instant-NGP [44] | 26.79 | 24.76 | 24.27 | 24.27 | 25.02 | 0.746 | 0.639 | 0.626 | 0.698 | 0.677 | 0.239 | 0.367 | 0.445 | 0.475 | 0.382 |
| Mip-NeRF 360 [2] | 29.26 | 25.18 | 24.16 | 24.10 | 25.67 | 0.860 | 0.727 | 0.670 | 0.706 | 0.741 | 0.122 | 0.260 | 0.370 | 0.428 | 0.295 |
| Zip-NeRF [3] | 29.66 | 23.27 | 20.87 | 20.27 | 23.52 | 0.875 | 0.696 | 0.565 | 0.559 | 0.674 | 0.097 | 0.257 | 0.421 | 0.494 | 0.318 |
| 3DGS [31] | 29.19 | 23.50 | 20.71 | 19.59 | 23.25 | 0.880 | 0.740 | 0.619 | 0.619 | 0.715 | 0.107 | 0.243 | 0.394 | 0.476 | 0.305 |
| 3DGS [31] + EWA [82] | 29.30 | 25.90 | 23.70 | 22.81 | 25.43 | 0.880 | 0.775 | 0.667 | 0.643 | 0.741 | 0.114 | 0.236 | 0.369 | 0.449 | 0.292 |
| Mip-Splatting [75] | 29.39 | 27.39 | 26.47 | 26.22 | 27.37 | 0.884 | 0.808 | 0.754 | 0.765 | 0.803 | 0.108 | 0.205 | 0.305 | 0.392 | 0.252 |
| 2DGS | 28.82 | 24.97 | 23.79 | 23.55 | 25.28 | 0.869 | 0.755 | 0.691 | 0.713 | 0.757 | 0.118 | 0.251 | 0.367 | 0.435 | 0.293 |
| 2DGS w/o Clamping | 28.49 | 26.68 | 25.85 | 25.64 | 26.66 | 0.855 | 0.771 | 0.714 | 0.729 | 0.767 | 0.128 | 0.241 | 0.347 | 0.421 | 0.284 |
| AA-2DGS (ours) | 29.30 | 27.16 | 26.10 | 25.77 | 27.08 | 0.877 | 0.795 | 0.732 | 0.735 | 0.785 | 0.111 | 0.215 | 0.329 | 0.411 | 0.266 |

Table 3: **Single-scale Training and Multi-scale Testing on the Mip-NeRF 360 Dataset [2].** All methods are trained on the smallest scale ($1\times$) and evaluated across four scales ($1\times$, $2\times$, $4\times$, and $8\times$), with evaluations at higher sampling rates simulating zoom-in effects. Ours method significantly improves on the baseline 2DGS method across all scales even on the training resolution while having competitive results to Mip-Splatting.

**Single-scale Training and Same-scale Testing**  We perform here the standard benchmark evaluation on the Mip-NeRF 360 dataset [2], where models are trained and tested at the same resolution. Indoor scenes are downsampled by a factor of 2 and outdoor by 4. Table 5 shows that 3DGS based methods perform slightly better than the 2DGS based counterparts in this setting, where our method is still comparable to the baseline 2DGS method. Note that antialiasing methods like ours involve an inherent

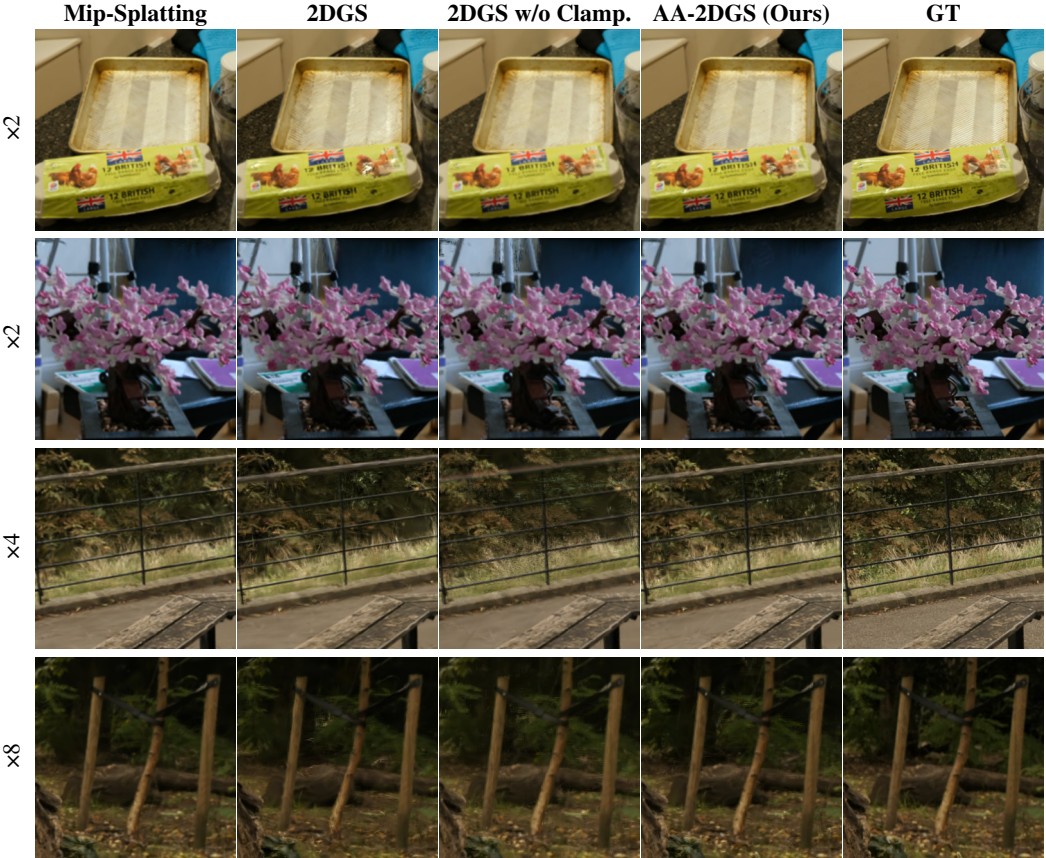

| | Mip-Splatting | 2DGS | 2DGS w/o Clamp. | AA-2DGS (Ours) | GT |

Figure 4: **Single-scale Training and Multi-scale Testing on Mip-NeRF 360 dataset [2]** All models are trained on 1/8 resolution and tested at different upscaling factors. Our AA-2DGS method maintains high fidelity when rendering at resolutions higher than the training resolution, reducing magnification artifacts compared to the baseline 2DGS method.

trade-off: by band-limiting the representation to prevent aliasing artifacts, we necessarily attenuate some high-frequency content. This can manifest as a small decrease in peak sharpness even at the original training resolution, resulting in slightly lower PSNR compared to the non-antialiased baseline method which is a fundamental trade-off between aliasing and sharpness. The minor reduction in single-scale PSNR is vastly outweighed by the significant improvements in non-training scale rendering, as demonstrated in the previous experiments.

| | | 24 | 37 | 40 | 55 | 63 | 65 | 69 | 83 | 97 | 105 | 106 | 110 | 114 | 118 | 122 | Mean |
|---|---|---|---|---|---|---|---|---|---|---|---|---|---|---|---|---|---|
| implicit | NeRF [43] | 1.90 | 1.60 | 1.85 | 0.58 | 2.28 | 1.27 | 1.47 | 1.67 | 2.05 | 1.07 | 0.88 | 2.53 | 1.06 | 1.15 | 0.96 | 1.49 |
| | VolSDF [70] | 1.14 | 1.26 | 0.81 | 0.49 | 1.25 | 0.70 | 0.72 | 1.29 | 1.18 | 0.70 | 0.66 | 1.08 | 0.42 | 0.61 | 0.55 | 0.86 |
| | NeuS [62] | 1.00 | 1.37 | 0.93 | 0.43 | 1.10 | 0.65 | 0.57 | 1.48 | 1.09 | 0.83 | 0.52 | 1.20 | 0.35 | 0.49 | 0.54 | 0.84 |
| | Neuralangelo [38] | 0.37 | 0.72 | 0.35 | 0.35 | 0.87 | 0.54 | 0.53 | 1.29 | 0.97 | 0.73 | 0.47 | 0.74 | 0.32 | 0.41 | 0.43 | 0.61 |
| explicit | 3DGS [31] | 2.14 | 1.53 | 2.08 | 1.68 | 3.49 | 2.21 | 1.43 | 2.07 | 2.22 | 1.75 | 1.79 | 2.55 | 1.53 | 1.52 | 1.50 | 1.96 |
| | SuGaR [19] | 1.47 | 1.33 | 1.13 | 0.61 | 2.25 | 1.71 | 1.15 | 1.63 | 1.62 | 1.07 | 0.79 | 2.45 | 0.98 | 0.88 | 0.79 | 1.33 |
| | GaussianSurfels [10] | 0.66 | 0.93 | 0.54 | 0.41 | 1.06 | 1.14 | 0.85 | 1.29 | 1.53 | 0.79 | 0.82 | 1.58 | 0.45 | 0.66 | 0.53 | 0.88 |
| | GOF [76] | 0.50 | 0.82 | 0.37 | 0.37 | 1.12 | 0.74 | 0.73 | 1.18 | 1.29 | 0.68 | 0.77 | 0.90 | 0.42 | 0.66 | 0.49 | 0.74 |
| | 2DGS [23] | 0.48 | 0.91 | 0.39 | 0.39 | 1.01 | 0.83 | 0.81 | 1.36 | 1.27 | 0.76 | 0.70 | 1.40 | 0.40 | 0.76 | 0.52 | 0.80 |
| | 2DGS* [23] | 0.50 | 0.77 | 0.36 | 0.36 | 0.91 | 0.81 | 0.78 | 1.26 | 1.22 | 0.67 | 0.68 | 1.26 | 0.38 | 0.85 | 0.49 | 0.76 |
| | 2DGS w/o Clamping | 0.49 | 0.80 | 0.33 | 0.36 | 0.96 | 0.89 | 0.78 | 1.30 | 1.24 | 0.67 | 0.66 | 1.33 | 0.37 | 0.65 | 0.45 | 0.75 |
| | AA-2DGS (ours) | 0.49 | 0.77 | 0.35 | 0.37 | 0.87 | 0.83 | 0.78 | 1.25 | 1.23 | 0.66 | 0.71 | 1.19 | 0.38 | 0.69 | 0.47 | 0.74 |

Table 4: **Quantitative comparison on the DTU Dataset [29].** We use reported Chamfer distance results from [76]. ∗ indicates that we retrain the model.

| | PSNR ↑ | SSIM ↑ | LPIPS ↓ |
|---|---|---|---|
| NeRF [43, 11] | 23.85 | 0.605 | 0.451 |
| mip-NeRF [1] | 24.04 | 0.616 | 0.441 |
| NeRF++ [77] | 25.11 | 0.676 | 0.375 |
| Plenoxels [15] | 23.08 | 0.626 | 0.463 |
| Instant NGP [44] | 25.68 | 0.705 | 0.302 |
| mip-NeRF 360 [2] | 27.57 | 0.793 | 0.234 |
| Zip-NeRF [3] | 28.54 | 0.828 | 0.189 |
| 3DGS [31] | 27.21 | 0.815 | 0.214 |
| 3DGS [31] + EWA [82] | 27.77 | 0.826 | 0.206 |
| Mip-Splatting [75] | 27.79 | 0.827 | 0.203 |
| 2DGS [23] | 27.56 | 0.819 | 0.209 |
| 2DGS w/o Clamping | 27.29 | 0.802 | 0.232 |
| AA-2DGS (ours) | 27.38 | 0.816 | 0.216 |

Table 5: **Single-scale Training and Same-scale Testing on the Mip-NeRF 360 dataset [2].** In the standard in-distribution setting, our approach still demonstrates performance on par with the baseline 2DGS method.

## 4.4 Evaluation on the DTU Dataset

The DTU dataset [29] counts 15 scenes, each with 49 or 69 images. We use downsampled images to 800×600. We follow previous methods [76, 23] for this evaluation. We report reconstruction performances in Table 4. NeuralAngelo [38] is among the state-of-the-art methods in this benchmark. However, Such implicit methods can be very slow to train, taking more than 12 hours at times on standard GPUs. The 3DGS representation evidently fails to recover meaningful depth despite good

novel view synthesis performance. AA-2DGS, 2DGS and 2DGS w/o Clamping perform almost similarly, with a slight edge in favor of our method. This shows that our anti-aliasing mechanisms integration within the 2DGS representation preserves its geometric modelling capabilities. The performance we obtain is on par with recent stat-of-the-art Gaussian Splatting based reconstruction methods. Additionally, we note that the benefits of our anti-aliasing method are not confined to RGB output, but naturally extend to all rendered attributes as shown through normal rendering in figure 5. This can improve accuracy in applications like surface reconstruction and reflective scene modelling, especially when multiscale input images are used for the training.

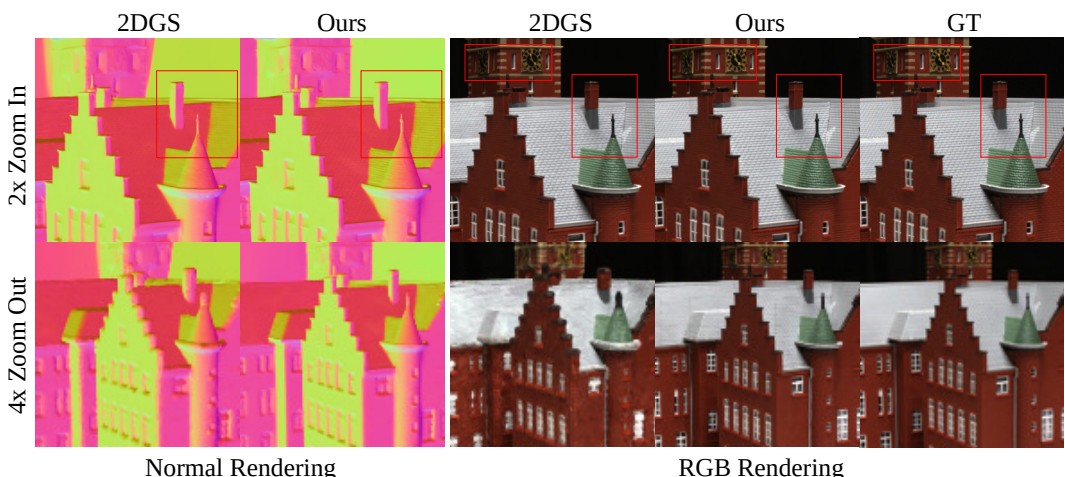

Figure 5: 2DGS and our method's RGB and normal rendering under different image sampling rates than the training views. We show results of simulating Zoom In (2x) and Zoom Out (4x). In addition to anti-aliased color rendering, our method also improves other attributes rendering.

## 5 Limitations

While our method significantly reduces aliasing in 2D Gaussian Splatting, it is not without limitations. A fundamental issue stems from the planar nature of 2D Gaussians, which can still produce "needle-like" artifacts in magnification scenarios or in extreme grazing angles viewing. Our world-space smoothing mitigates this by enforcing a minimum screen footprint, but it cannot fundamentally solve the zero-thickness problem in the direction normal to the primitive.

Furthermore, our approach involves a classic trade-off between antialiasing and detail preservation. The fixed filter parameters, while effective in general, may not be optimal for all scenes and can lead to over-smoothing. A more detailed analysis of these limitations, is provided in Appendix C.

## 6 Conclusion

We introduced Anti-Aliased 2D Gaussian Splatting (AA-2DGS), a method that enables high-quality antialiasing for 2D Gaussian primitives while preserving their geometric accuracy. Our approach combines a world-space flat smoothing kernel that constrains the frequency content of 2D Gaussian primitives based on training view sampling rates, and an object-space Mip filter that leverages the ray-splat intersection mapping to perform prefiltering directly in the local space of each splat. By incorporating these techniques, AA-2DGS effectively mitigates aliasing artifacts when rendering at different sampling rates. Our experiments demonstrate that AA-2DGS consistently outperforms the original 2DGS method across standard novel view synthesis benchmarks for varied sampling rates and mixed resolution training while maintaining mesh reconstruction capabilities. This work bridges the gap between the geometric accuracy of 2DGS and the high-quality antialiasing capabilities previously only available to volumetric 3D Gaussian representations, enabling more robust and visually pleasing results in applications requiring precise geometry.

**Potential Societal Impact**    We do not identify any specific societal risks that require special attention within the scope of this work.

**Acknowledgment** This work was granted access to the HPC resources of IDRIS under the allocation 20XX-AD010616156 made by GENCI.

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

## A Ablation Studies

### A.1 Effectiveness of the World-Space Flat Smoothing Kernel

| | PSNR ↑ | | | | | SSIM ↑ | | | | | LPIPS ↓ | | | | |
|---|---|---|---|---|---|---|---|---|---|---|---|---|---|---|---|
| | 1× Res. | 2× Res. | 4× Res. | 8× Res. | Avg. | 1× Res. | 2× Res. | 4× Res. | 8× Res. | Avg. | 1× Res. | 2× Res. | 4× Res. | 8× Res. | Avg. |
| 2DGS [23] | 28.82 | 24.97 | 23.79 | 23.55 | 25.28 | 0.869 | 0.755 | 0.691 | 0.713 | 0.757 | 0.118 | 0.251 | 0.367 | 0.435 | 0.293 |
| 2DGS w/o Clamping | 28.49 | 26.68 | 25.85 | 25.64 | 26.66 | 0.855 | 0.771 | 0.714 | 0.729 | 0.767 | 0.128 | 0.241 | 0.347 | 0.421 | 0.284 |
| AA-2DGS (ours) | 29.30 | 27.16 | 26.10 | 25.77 | 27.08 | 0.877 | 0.795 | 0.732 | 0.735 | 0.785 | 0.111 | 0.215 | 0.329 | 0.411 | 0.266 |
| AA-2DGS (ours) w/o flat smoothing filter | 29.09 | 26.85 | 25.63 | 25.18 | 26.69 | 0.875 | 0.788 | 0.716 | 0.709 | 0.772 | 0.115 | 0.226 | 0.352 | 0.434 | 0.282 |
| AA-2DGS (ours) w/o Mip filter | 28.75 | 26.85 | 26.00 | 25.78 | 26.85 | 0.862 | 0.777 | 0.722 | 0.738 | 0.775 | 0.122 | 0.233 | 0.338 | 0.415 | 0.277 |

Table 6: **Single-scale Training and Multi-scale Testing on the Mip-NeRF 360 Dataset [2].** All methods are trained on the smallest scale (1×), corresponding to eighth of the original image resolution, and evaluated across four scales (1×, 2×, 4×, and 8×), with evaluations at higher sampling rates simulating zoom-in effects. Disabling the world-space flat smoothing filter results in high-frequency magnification artifacts when rendering higher resolution images. Disabling the 2D Mip filter causes a slight decline in performance at high magnification.

In order to assess the effectiveness of the World-Space Flat Smoothing Kernel, we show an ablation with an experiment on the single-scale training and multi-scale testing setting in the Mip-NeRF 360 dataset [2] to simulate magnification or Zoom In effects. We present quantitative results in Table 6. It shows that performance degrades at higher resolution than the training one when disabling the flat smooth kernel due to high-frequency magnification artifacts.

In this experiment, the Object-Space Mip filter mostly improves results at the training resolution and does not improve much at higher ones because it is primarily designed to address aliasing in minification scenarios as we show in A.2.

However, for magnification, where the rendering sampling rate exceeds the frequency content available in the trained representation, this additional filtering can sometimes lead to over-smoothing of details that would naturally become visible when zooming in, especially at extreme magnifications.

2D Gaussians are fundamentally planar primitives with zero thickness orthogonal to their surface. When viewed from grazing angles, they project to extremely thin lines on the screen, creating "needle-like" artifacts. This is particularly problematic during magnification, as primitives optimized for lower resolutions suddenly reveal their orientation-dependent thinness. The flat smoothing kernel helps mitigate this issue by ensuring a minimum footprint size in the tangent plane, but cannot address the fundamental zero-thickness property in the normal direction.

In contrast, 3D Gaussians in Mip-Splatting are volumetric primitives that maintain substantial screen presence even from oblique viewpoints. Their three-dimensional nature allows the 3D smoothing kernel to effectively regularize their shape in all directions, leading to more consistent results across viewing angles and scales.

Despite these inherent limitations of planar primitives, our method still demonstrates meaningful improvements over the original 2DGS approach. As shown in Table 6, the combination of World-Space Flat Smoothing Kernel and Object-Space Mip Filter consistently outperforms both the clamping-based approach of the original 2DGS and the non-clamped variant.

### A.2 Effectiveness of the Object-Space Mip Filter

To evaluate the effectiveness of the Object-Space Mip filter, we perform an ablation study with the single-scale training and multi-scale testing setting to simulate zoom-out effects in the Blender dataset [43]. Quantitative results are shown in Table 7. Similar to previous experiments, we find that disabling the clamping heuristic performed by 2DGS [23] (*2DGS w/o Clamping*), the dilation artifacts are eliminated, outperforming vanilla 2DGS. However, it still shows severe aliasing artifacts especially at extreme zoom out. AA-2DGS outperforms all 2DGS variants by a large gap in this

| | PSNR ↑ | | | | | SSIM ↑ | | | | | LPIPS ↓ | | | | |
|---|---|---|---|---|---|---|---|---|---|---|---|---|---|---|---|
| | Full Res. | 1/2 Res. | 1/4 Res. | 1/8 Res. | Avg. | Full Res. | 1/2 Res. | 1/4 Res. | 1/8 Res. | Avg. | Full Res. | 1/2 Res. | 1/4 Res. | 1/8 Res | Avg. |
| 2DGS [23] | 33.05 | 27.64 | 20.61 | 16.59 | 24.47 | 0.967 | 0.952 | 0.856 | 0.720 | 0.874 | 0.033 | 0.037 | 0.082 | 0.151 | 0.076 |
| 2DGS w/o Clamping | 33.18 | 33.04 | 29.74 | 26.21 | 30.54 | 0.968 | 0.973 | 0.964 | 0.945 | 0.963 | 0.032 | 0.024 | 0.040 | 0.074 | 0.042 |
| AA-2DGS (ours) | 33.24 | 34.10 | 32.11 | 29.00 | 32.11 | 0.967 | 0.976 | 0.978 | 0.973 | 0.974 | 0.034 | 0.020 | 0.019 | 0.024 | 0.024 |
| AA-2DGS (ours) w/o flat smoothing filter | 33.38 | 34.08 | 31.96 | 28.84 | 32.06 | 0.968 | 0.976 | 0.978 | 0.973 | 0.973 | 0.033 | 0.020 | 0.019 | 0.024 | 0.024 |
| AA-2DGS (ours) w/o Mip filter | 33.13 | 33.51 | 30.06 | 26.36 | 30.76 | 0.967 | 0.974 | 0.966 | 0.946 | 0.963 | 0.033 | 0.022 | 0.038 | 0.072 | 0.041 |

Table 7: **Single-scale Training and Multi-scale Testing on the Blender Dataset [43].** All methods are trained on full-resolution images and evaluated at four different (smaller) resolutions, with lower resolutions simulating minification / zoom-out effects. Our method achieves results that are comparable at training resolution to 2DGS methods while significantly surpassing them at lower scales. When disabling the Object-Space Mip filter, we obtain worse results at lower scales, which shows its effectiveness in this experiment. On the other hand, disabling the world-space flat smoothing filter leads to mostly similar performance since it is more involved in handling magnification artifacts.

experiment. Disabling the Object-Space Mip filter results in noticeable degradation in performance, validating its important role in anti-aliasing in this minification experiment. Without the world-space flat smoothing filter, our method still produces anti-aliased rendering as the smoothing filter is designed to tackle high-frequency artifacts during magnification as shown previously.

# B    Additional Qualitative Results

## B.1    Additional Results for Single-scale Training and Multi-scale Testing on the Blender Dataset

In this section, we show additional qualitative results in Figure 6 for the minification/ Zoom Out experiments of Single-scale Training and Multi-scale Testing on the Blender Dataset [43].

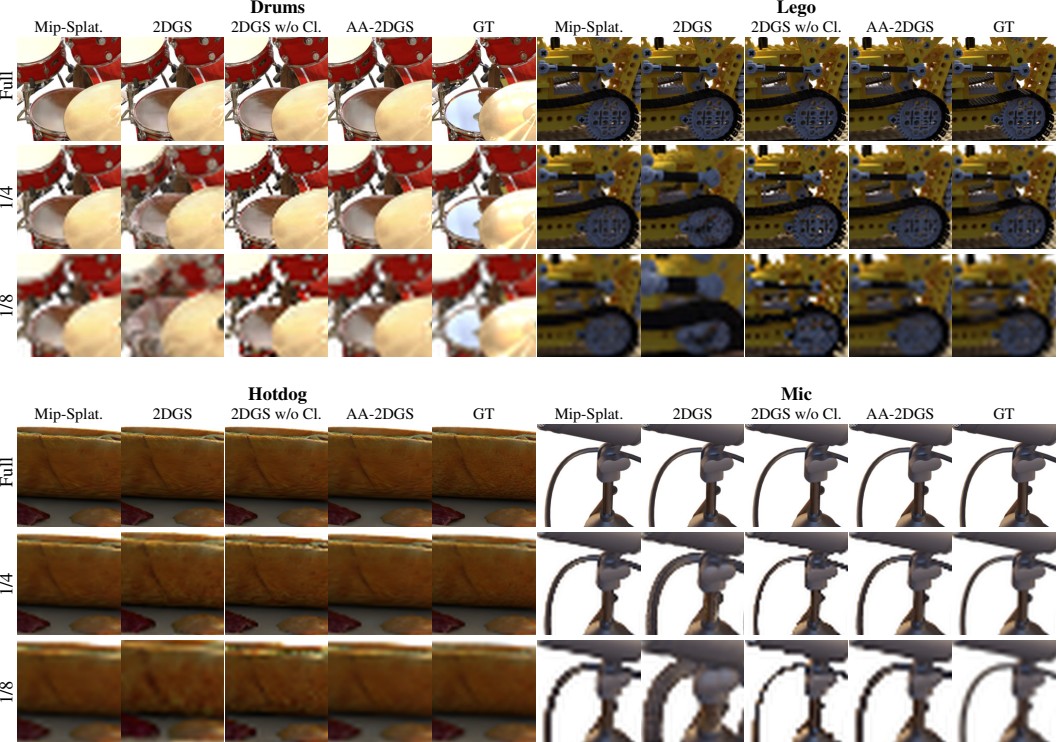

Figure 6: **Additional Results of Single-scale Training and Multi-scale Testing on the Blender Dataset [43].** All methods are trained at full resolution and evaluated at smaller resolutions to simulate Zoom Out/ magnification. Our method (AA-2DGS) consistently demonstrates improved quality across all sampling rates compared to the baseline 2DGS method.

## B.2    Additional Results for Single-scale Training and Multi-scale Testing on the Mip-NeRF 360 Dataset

In this section, we show additional qualitative results in Figure 7 for the magnification/ Zoom In experiments of Single-scale Training and Multi-scale Testing on the Mip-NeRF 360 Dataset [2].

| Mip-Splatting | 2DGS | 2DGS w/o Clamp. | AA-2DGS (Ours) | GT |
| --- | --- | --- | --- | --- |

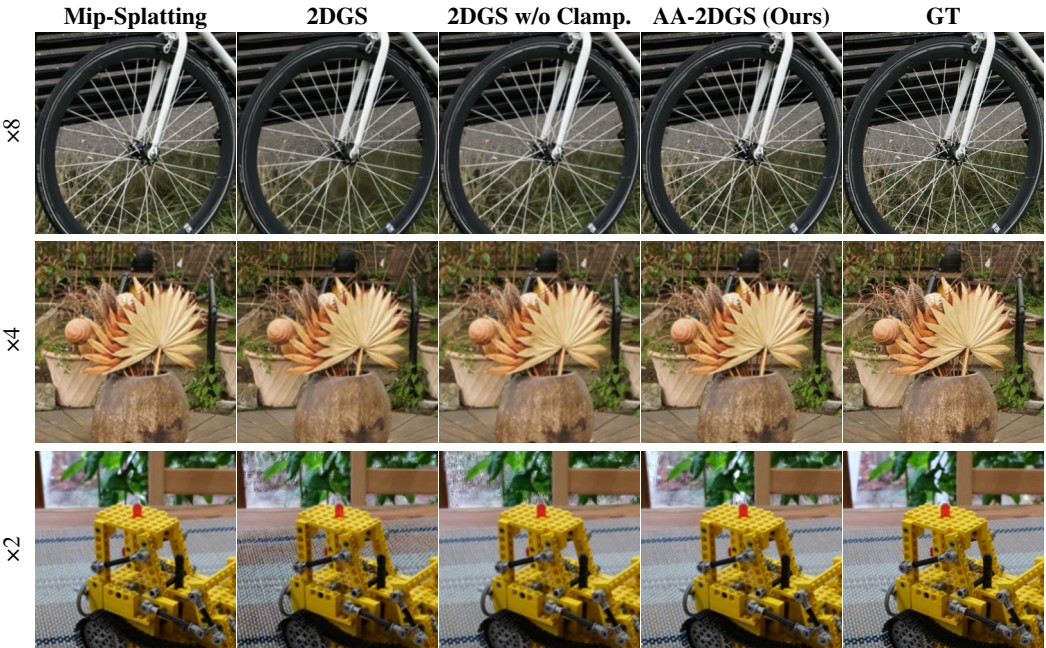

Figure 7: **Additional Results of Single-scale Training and Multi-scale Testing on the Mip-NeRF 360 Dataset.** All models are trained on 1/8 resolution and tested at different upscaling factors: Bicycle (×8), Garden (×4), and Bonsai (×2). Our AA-2DGS method maintains high fidelity when rendering at resolutions significantly higher than the training resolution, reducing artifacts compared to the baseline methods.

## C   Detailed Discussion on Limitations

While our antialiasing approach for 2D Gaussian Splatting demonstrates significant improvements over the original implementation, it is important to acknowledge certain limitations.

**Balancing Antialiasing and Detail Preservation**   Like all antialiasing techniques, our method faces an inherent trade-off between removing aliasing artifacts and preserving fine details. We use the same filter values as Mip-Splatting: $\sigma = 0.1$ for the Object-Space Mip Filter and $s = 0.2$ for the World-Space Flat Smoothing Kernel. While these values provide a good balance for most scenes, optimal parameters may vary across different datasets or viewing conditions.

**Inherent Limitations of Planar Primitives**   A fundamental limitation stems from the nature of 2D Gaussians as planar primitives with zero thickness orthogonal to their surface. As shown in our ablation studies (Table 6), when viewed from grazing angles, these primitives project to extremely thin lines on the screen, creating "needle-like" artifacts. This is particularly problematic during extreme magnification, as primitives optimized for lower resolutions suddenly reveal their orientation-dependent thinness.

The World-Space Flat Smoothing Kernel helps mitigate this issue by ensuring a minimum footprint size in the tangent plane, but cannot address the fundamental zero-thickness property in the normal direction. In contrast, volumetric primitives like those used in 3D Gaussian Splatting maintain substantial screen presence even from oblique viewpoints, allowing their smoothing kernels to regularize shape in all directions.

**Magnification and Minification Trade-offs**   As demonstrated in our ablation studies, the Object-Space Mip Filter primarily addresses aliasing in minification scenarios (zoom-out), while the World-Space Flat Smoothing Kernel targets high-frequency artifacts during magnification (zoom-in). For extreme magnification cases where the rendering sampling rate exceeds the frequency content available in the trained representation, our filtering approach can sometimes lead to over-smoothing of details that would naturally become visible when zooming in.

Despite these limitations, our experiments consistently show that the combination of World-Space Flat Smoothing Kernel and Object-Space Mip Filter outperforms both the clamping-based approach

of the original 2DGS and non-clamped variants, particularly in challenging multi-scale rendering scenarios. Our method provides a more principled approach to antialiasing for 2D Gaussian Splatting while maintaining the computational efficiency and view-consistent geometry that makes 2DGS attractive.

