# OpenReview forum: "Anti-Aliased 2D Gaussian Splatting"
_NeurIPS.cc/2025/Conference — NeurIPS 2025 poster_

### Official Review · Reviewer_EnMS · 2025-07-01

**Clarity:** 3
**Significance:** 2
**Originality:** 2
**Rating:** 3
**Confidence:** 4

**Summary:**

In this paper, the authors introduce Anti-Aliased 2D Gaussian Splatting (AA-2DGS) by identifying two key insights: 1) first, they impose a **world-space flat smoothing kernel** that constrains the frequency content of each 2D Gaussian primitive based on the maximal sampling rate of the training views, effectively eliminating high-frequency artifacts when zooming in; 2) second, they derive an **object-space Mip filter** via an affine approximation of the ray-splat intersection mapping, allowing proper anti-aliasing to be applied directly in each splat’s local coordinate frame.

By integrating these mechanisms, AA-2DGS mitigates aliasing under both zoom-in and zoom-out conditions while preserving the precise geometry that makes 2DGS valuable. Experimental results show that AA-2DGS consistently outperforms vanilla 2DGS and rivals state-of-the-art anti-aliased 3DGS methods in PSNR, SSIM, and LPIPS across all resolutions.

**Questions:**

- In Table 5, I notice comparisons with Zip-NeRF and Mip-NeRF 360. However, those two methods use different image resolutions than vanilla 3D-GS (Mip-NeRF 360 downsamples outdoor scenes by $4x$ and indoor scenes by $2x$, whereas vanilla 3D-GS uniformly crops the width to 1600 pixels). Did the authors adopt the same resolution settings across all methods? Such differences can significantly impact the reported rendering metrics.

**Ethical Concerns:**

["NO or VERY MINOR ethics concerns only"]

**Final Justification:**

I still feel that the authors somewhat overstate the significance of their contributions. At its core, the work primarily involves adapting mip-splatting to 2DGS, with the necessary modifications for that transition. Additionally, I remain unconvinced by the motivation behind AA-2DGS, as I believe anti-aliasing and geometry reconstruction are fundamentally unrelated. The experimental results further suggest that AA-2DGS performs worse in multi-scale rendering compared to 2DGS and shows inferior geometry reconstruction compared to state-of-the-art methods based on 2DGS. While I will raise the score based on the authors’ rebuttal, I still lean toward a weak rejection. However, I am open to accepting the paper if deemed appropriate.

**Limitations:**

Yes.

**Paper Formatting Concerns:**

N/A.

**Quality:**

3

**Strengths And Weaknesses:**

Strengths:

- This paper is well-written and easy to follow.
- Decent results for anti-aliasing and geometry reconstruction.

Weaknesses:

- Missing references for dynamic reconstruction based on 3D-GS:

  - Real-time Photorealistic Dynamic Scene Representation and Rendering with 4D Gaussian Splatting, by Zeyu Yang et al. ICLR 2024. The first 4D Gaussian splatting.
  - Deformable 3D Gaussians for High-Fidelity Monocular Dynamic Scene Reconstruction, by Ziyi Yang et al. CVPR 2024. The first deformable Gaussian splatting.

  AbsGS (AbsGS: Recovering Fine Details for 3D Gaussian Splatting) should also be cited under the methods to improve density control in Line 91.

- **Lack of novelty**. The paper’s contribution is largely one of adaptation rather than innovation. It essentially ports Mip-Splatting to the 2D-GS framework without proposing new smoothing strategies. The authors transform the 3D Gaussian low-pass filter into a planar, world-space “flat” smoothing kernel and change the screen-space Mip filter into each splat’s local coordinate frame.

- I find the stated purpose of AA-2DGS somewhat unclear. Had the authors concentrated their contributions on improving the mesh quality produced by 2D Gaussian Splatting, I would rate the paper more highly. Instead, they place the bulk of their emphasis on anti-aliasing. It remains unclear what inherent advantage 2DGS holds over 3DGS in addressing aliasing: after all, 2DGS is merely a specialized case of Gaussian splatting tailored for geometry reconstruction, and subsequent works have shown that for pure rendering tasks, 2DGS offers no substantial benefit over its 3D counterpart.

- **Missing qualitative results on mesh.** The Experiments section omits any qualitative visualizations of the reconstructed mesh. Since AA-2DGS is fundamentally an anti-aliasing extension of 2DGS, the paper would benefit from showing rendered mesh outputs to illustrate geometric improvements. As it stands, only Table 4 reports reconstruction metrics, with no accompanying mesh visual examples.

- Missing comparison on FPS.

---

> ### Author Rebuttal · Authors · 2025-07-30
>
> We thank the reviewer for their time and feedback. We appreciate their positive comments on the paper's clarity, results, and organization. We will correct any missing references, formatting issues, or other minor points in the final version of the paper.
>
> We would like to address the main weaknesses and questions raised, as we believe they stem from a few key misunderstandings of our work's motivation and technical novelty.
>
> ### On the Purpose and Significance of Anti-Aliasing for 2DGS
>
> The reviewer questions the purpose of our work, suggesting we should have focused on improving mesh quality and that "2DGS offers no substantial benefit over its 3D counterpart" for rendering. We are afraid this reflects a misunderstanding of our central motivation.
>
> We would like to clarify a key point regarding the paper's goal for geometry, which appears to have been misinterpreted. The central claim is one of **preservation, not improvement**.
>
> Our work is motivated by the fact that 2DGS possesses established geometric advantages over 3DGS, but its rendering limitations due to aliasing hinder its wider practical use and adoption. Therefore, our primary objective was to design an anti-aliasing solution that could be integrated into the pipeline without degrading this geometric fidelity. The DTU experiment (Table 4) was undertaken specifically to validate this claim of preservation. The results, showing our method performing on par with the strong baseline, should thus be interpreted as a success: we fixed the rendering problems while maintaining the geometric strengths.
>
> 2DGS is a vital and active research direction precisely because of its geometric accuracy. As we state in the introduction, 2DGS provides superior depth and normal reconstruction compared to 3DGS. This has led to a recent surge in follow-up work that builds upon the 2DGS representation for tasks where geometry is paramount and integral to rendering, such as in reflective scene modeling or physics based rendering where rendering is dependent on reflection direction (ie normal), e.g.:
> *   *Ref-GS: Directional Factorization for 2D Gaussian Splatting*. CVPR 25
> *   *Ref-Gaussian Reflective Gaussian Splatting*. ICLR 25
>
> , sparse-view joint reconstruction and novel view synthesis, e.g.:
> *   *MAtCha Gaussians: Atlas of Charts for High-Quality Geometry and Photorealism From Sparse Views*, CVPR 25
>
> , physics-based inverse rendering with ray tracing, e.g.:
> *   *IRGS: Inter-Reflective Gaussian Splatting with 2D Gaussian Ray Tracing*, CVPR 25
>
> An anti-aliased 2DGS representation is highly impactful and beneficial for all such applications of 2DGS.
>
> *   **The primary limitation of 2DGS is aliasing.** Its inability to render well at different scales (e.g., camera zoom) prevents its widespread adoption despite its geometric advantages. Our work directly addresses this bottleneck.
> *   **Our aim is a unified representation.** We seek to create a single, robust representation that excels at both high-quality, multi-scale rendering and accurate surface reconstruction from a single training, combining the best of both worlds. The experiments were therefore designed to show that we significantly improve rendering while maintaining the state-of-the-art geometric performance of 2DGS.
>
> This was also agreed on by other reviewers:
> > **MET9** “This directly addresses a critical limitation of 2DGS, enhancing its practical applicability.”
>
> > **oTXT** “The paper is well-written and addresses an important problem: extending anti-aliasing to 2D Gaussian Splatting representations. This has the potential for significant impact given the widespread use of 2D GS.”
>
> > **4K14** “The motivation and solution is clear. 2DGS has aliasing issues due to max operator”
>
> ### On Novelty: Beyond a Simple Adaptation of Mip-Splatting
>
> The reviewer characterizes our work as a simple adaptation of Mip-Splatting. We respectfully disagree and wish to highlight our novel technical contributions. Our framework consists of two distinct components, each addressing a different aspect of aliasing with a different level of innovation.
>
> **1. A Novel Framework is Required for the 2DGS Rendering Pipeline.**
> The rendering mechanisms of 3DGS and 2DGS are fundamentally incompatible, precluding any direct application of 3DGS-based solutions.
> *   **3DGS** projects 3D Gaussians to screen space, which is a well-defined affine transformation. Applying screen-space EWA filtering is a direct extension of existing literature.
> *   **2DGS** does not project primitives. Instead, it computes a complex, non-linear ray-splat intersection (Eq. 5 and 6) to find a point in the planar splat’s local 2D space.
>
> It is not at all trivial how to perform frequency-based band-limiting or pre-filtering in this context. Our "Object-Space Mip Filter" is a novel solution tailored specifically for this unique intersection-based rendering paradigm. **It is a new mathematical derivation with broader impact.**
>
> Our technical approach is fundamentally different and more novel than that of Mip-Splatting. Mip-Splatting uses the standard screen-space EWA formulation and simply modifies the filter kernel size.
>
> In contrast, our work introduces a new mathematical derivation for an anti-aliasing filter that operates directly in the object space of each splat.
> The derivation proceeds as follows:
> *   **The Mapping:** We begin with the ray-splat intersection function $m(\mathbf{x}) = \mathbf{u}(\mathbf{x})$ on the splat (Eq. 5 and 6).
> *   **The Core Innovation:** We form a first-order Taylor approximation of this complex mapping around a pixel, defined by its Jacobian $\mathbf{J} = \frac{\partial \mathbf{u}}{\partial \mathbf{x}}$ (Eq. 12). This local affine approximation is the key insight that makes an analytical solution possible.
> *   **Mapping to Screen Space:** Using this affine mapping and a fundamental property of Gaussian functions, we can express how the local unit Gaussian behaves in screen space (Eq. 13).
> *   **Filtering in Screen Space:** We then mathematically convolve this transformed Gaussian with a screen-space Mip filter (a Gaussian with covariance $\sigma\mathbf{I}$) to obtain a filtered Gaussian with covariance $\mathbf{J}^{-1}\mathbf{J}^{-\top} + \sigma\mathbf{I}$ (Eq. 15).
> *   **Mapping Back to Object Space:** Finally, we use the same affine transform properties to map this filtered screen-space Gaussian back into the splat's local `uv`-space. This yields our final, novel result: the primitive can be evaluated as a single Gaussian in its local space with a new, dynamically computed covariance $\boldsymbol{\Sigma}_{\text{local}}' = \mathbf{I} + \sigma\mathbf{J}\mathbf{J}^\top$ (Eq. 17).
>
> To our knowledge, this formulation—pre-filtering an intersection-based primitive by analytically deriving an equivalent object-space covariance from a screen-space filter—is a new contribution that has not been proposed before, not even in classical graphics literature on **surface splatting**, highlighting its novelty and potential impact beyond just 2DGS.
>
> **2. The World-Space Flat Smoothing Kernel: A Necessary and Non-Trivial Formulation for Planar Primitives.**
>
> Our second component addresses pre-aliasing by band-limiting the representation, conceptually inspired by the frequency analysis in Mip-Splatting. However, a 3D volumetric filter cannot be applied to a 2D planar primitive. Our contribution was to correctly formulate this regularization for a flattened representation by projecting the 3D isotropic smoothing kernel onto the 2D splat plane. This results in our effective planar covariance matrix ($\mathbf{V}_k^{\text{eff}}$ in Eq. 9) and corresponding opacity modulation (Eq. 10), which is a necessary step to complete the anti-aliasing framework for 2DGS, especially for the case of magnification.
>
> **3. We provide new insights into the limitations of the original 2DGS.**
>
> A key contribution of our work is the analysis and demonstration of the clamping heuristic's ineffectiveness in the original 2DGS. Prior work had not identified this as a critical source of artifacts. Our experiments (2DGS w/o Clamping in Tables 1-5, and in the Appendix) provides the first concrete evidence that this heuristic often exacerbates aliasing and damages performance, especially during minification. This insight itself is a valuable contribution to the understanding of the vanilla 2DGS method.
>
> ### On Missing Qualitative Mesh Results & FPS Comparison
>
> **Mesh Visualizations:** Our initial focus on quantitative metrics in Table 4 was to rigorously demonstrate that our method preserves the state-of-the-art geometric accuracy of 2DGS, which was our central claim for this aspect of our work. We deemed numerical results sufficient as they clearly show preservation of baseline performance. To the request of the reviewer, we will add mesh visualizations.
>
> **FPS Comparison:** The reviewer states that a comparison on FPS is missing. **This is a factual error.** We explicitly report the overhead in Section 4.1, "Implementation Details":
> > "our approach incurs a slight overhead of 10-15% in rendering time compared to the original 2DGS implementation."
>
> ### On Experimental Settings (Table 5)
>
> The reviewer correctly notes that different methods on the Mip-NeRF 360 benchmark use specific resolution settings. We confirm we followed the standard protocol. As stated in Section 4.3 (Single-scale Training and Same-scale Testing):
> > "Indoor scenes are downsampled by a factor of 2 and outdoor by 4."
>
> The reported numbers for other methods are taken from Mip-Splatting, which used the same standard setup, thus ensuring a fair comparison.
>
> We hope these clarifications address the reviewer's concerns and underscore the novelty and significance of our contributions. We are confident that AA-2DGS is a valuable step forward in making a geometrically superior representation practical for high-quality, multi-scale rendering. We thank the reviewer again for their constructive feedback.

---

> > ### Comment · Reviewer_EnMS · 2025-08-07
> >
> > Thank you for the authors' detailed response. Most of my concerns have been addressed, but a few still remain. While I acknowledge that AA-2DGS is not merely a straightforward adaptation of Mip-Splatting to 2DGS, the core idea remains similar. As such, I still lean toward viewing AA-2DGS as more of an engineering effort rather than a fundamentally innovative work.
> >
> > Additionally, I respectfully disagree with the authors' claim that *"AA-2DGS is a valuable step forward in making a geometrically superior representation practical for high-quality, multiscale rendering."* In the field of neural rendering, rendering quality and geometry reconstruction often involve trade-offs—high-quality (and multiscale) rendering should not be conflated with high-fidelity geometry reconstruction. If the focus is solely on achieving high-quality, multiscale rendering within a geometric representation, the authors should further compare AA-2DGS with PGSR and RaDe-GS in Table 4 for a fair assessment.

---

> > > ### Author Response · Authors · 2025-08-08
> > >
> > > We thank the reviewer for the continued discussion and for acknowledging that our work is more than a straightforward adaptation. We would like to address the remaining points with further clarification.
> > >
> > > **1. On Innovation and a "Similar Core Idea" ?**
> > >
> > > While our work and Mip-Splatting share the high-level goal of anti-aliasing, we must respectfully emphasize that the core technical ideas and implementation strategies are fundamentally different, necessitated by the incompatible rendering pipelines of 2DGS and 3DGS.
> > >
> > > *   **Mip-Splatting** modifies the filter kernel size within the standard screen-space EWA framework, which is applicable because 3DGS involves a direct projection of primitives to the screen.
> > > *   **Our contribution** is the design of a novel **object-space filtering strategy**, a solution required because 2DGS uses a complex, non-linear ray-splat intersection instead of projection. This was achieved by developing a new formulation based on a **local affine approximation** of the screen-to-object mapping.
> > >
> > > This distinction is not minor; it represents a different conceptual approach to pre-filtering that is new to this rendering paradigm and the **proposed mathematical framework** is not a simple engineering effort.
> > >
> > > **2. On Unifying Rendering and Geometry: Achieving a Superior Trade-off**
> > >
> > > We agree with the reviewer that a trade-off between rendering quality and geometric fidelity often exists. Our work's central thesis is not to "conflate" these goals, but to demonstrate that it is possible to achieve a **superior and more practical trade-off** within a single, efficient model. Our claim is centered on the **multi-scale rendering**.
> > >
> > > This is not a hypothetical claim; our results provide direct empirical evidence. By comparing across our tables, one can see that:
> > > *   **AA-2DGS provides geometry that is significantly superior to 3DGS-based methods**.
> > > *   Simultaneously, **AA-2DGS delivers comparable state-of-the-art, multi-scale rendering performance to Mip-Splatting** (Tables 1, 2, and 3).
> > >
> > > This is all achieved in a single, fast training cycle. Our work thus provides a unified and practical solution that excels on both fronts, directly addressing the very trade-off the reviewer highlights and offering a better balance than prior methods. This has clear benefits for applications that require both high-fidelity geometry and robust, alias-free rendering, as we elaborated previously.
> > >
> > > **3. Regarding the Suggested Comparisons to PGSR and RaDe-GS**
> > >
> > >
> > > Comparing our fundamentally anti-aliasing method against geometric enhancement ones would not be an apples-to-apples comparison. Our contribution is the anti-aliasing framework itself, upon which such geometric regularizers could potentially be built in future work. We reiterate that the purpose of our DTU experiment was strictly to validate that our anti-aliasing solution preserves the strong geometric baseline of 2DGS, a claim which our results strongly support.

---

### Official Review · Reviewer_4K14 · 2025-07-01

**Clarity:** 3
**Significance:** 2
**Originality:** 2
**Rating:** 4
**Confidence:** 5

**Summary:**

This paper address the aliasing issues in 2D Gaussian Splatting. It adapted the idea from Mip-Splatting to 2DGS. Specifically, it projects the 3D smoothing filter in mip-splatting to the 2D Gaussian plane and apply the smoothing filter. It also unprojects the screen space 2D mip filter in mip-splatting to 2D Gaussian primitives' object space and apply the filter. The proposed method is valid in several datasets and shows anti-aliased results compared to 2DGS and the final results is comparable to mip-splatting.

**Questions:**

2DGS has surface reconstruction results on the tanks and temples datasets and I wonder if the anti-aliased solution also improve the resurface reconstruction those scenes.

**Ethical Concerns:**

["NO or VERY MINOR ethics concerns only"]

**Final Justification:**

After reading the rebuttal and other reviews, I think the paper nicely adapt the idea of mip-splatting to 2DGS to address the aliasing issues and I will maintain my rating as borderline accept.

**Limitations:**

I don't see significant limitations.

**Paper Formatting Concerns:**

Formatting looks good.

**Quality:**

3

**Strengths And Weaknesses:**

Strengths:
1. The motivation and solution is clear. 2DGS has aliasing issues due to max operator and mip-splatting's idea could be used to improve the results.
2. The writing is clear.
3. The proposed idea is validated in several datasets and experiments supported the claims.

Weakness:
1. The proposed idea is a straightforward adaptation of mip-splatting to 2DGS.
2. The results are not surprising as mip-splatting's solutions for anti-aliasing are very general.

---

> ### Author Rebuttal · Authors · 2025-07-30
>
> We sincerely thank the reviewer for their time and valuable feedback. We are pleased they found our paper’s motivation and writing to be clear and appreciate their positive assessment of our experimental validation. We would like to address the concerns raised regarding the work's originality and the scope of our evaluation to provide a clearer picture of our contributions.
>
> ### On the Novelty of our Method Beyond a "Straightforward Adaptation"
>
> We appreciate the reviewer's recognition that our work is inspired by Mip-Splatting. However, we wish to clarify that our method is not a "straightforward adaptation" but involves significant novel mathematical derivations necessitated by the fundamental rendering paradigm differences between 3DGS and 2DGS.
>
> **1. A Novel Framework is Required for the 2DGS Rendering Pipeline.**
> The rendering mechanisms of 3DGS and 2DGS are fundamentally incompatible, precluding any direct application of 3DGS-based solutions.
> *   **3DGS** projects 3D Gaussians to screen space, which is a well-defined affine transformation. Applying screen-space EWA filtering is a direct extension of existing literature.
> *   **2DGS** does not project primitives. Instead, it computes a complex, non-linear ray-splat intersection (Eq. 5 and 6) to find a point in the planar splat’s local 2D space.
>
> It is not at all trivial how to perform frequency-based band-limiting or pre-filtering in this context. Our "Object-Space Mip Filter" is a novel solution tailored specifically for this unique intersection-based rendering paradigm. **It is a new mathematical derivation with broader impact.**
>
> The reviewer notes that our method is "straightforward adaptation of mip-splatting". However, our technical approach is fundamentally different and more novel than that of Mip-Splatting. Mip-Splatting uses the standard screen-space EWA formulation and simply modifies the filter kernel size.
>
> In contrast, our work introduces a new mathematical derivation for an anti-aliasing filter that operates directly in the object space of each splat.
> The derivation proceeds as follows:
> *   **The Mapping:** We begin with the ray-splat intersection function $m(\mathbf{x}) = \mathbf{u}(\mathbf{x})$ on the splat (Eq. 5 and 6).
> *   **The Core Innovation:** We form a first-order Taylor approximation of this complex mapping around a pixel, defined by its Jacobian $\mathbf{J} = \frac{\partial \mathbf{u}}{\partial \mathbf{x}}$ (Eq. 12). This local affine approximation is the key insight that makes an analytical solution possible.
> *   **Mapping to Screen Space:** Using this affine mapping and a fundamental property of Gaussian functions, we can express how the local unit Gaussian behaves in screen space (Eq. 13).
> *   **Filtering in Screen Space:** We then mathematically convolve this transformed Gaussian with a screen-space Mip filter (a Gaussian with covariance $\sigma\mathbf{I}$) to obtain a filtered Gaussian with covariance $\mathbf{J}^{-1}\mathbf{J}^{-\top} + \sigma\mathbf{I}$ (Eq. 15).
> *   **Mapping Back to Object Space:** Finally, we use the same affine transform properties to map this filtered screen-space Gaussian back into the splat's local `uv`-space. This yields our final, novel result: the primitive can be evaluated as a single Gaussian in its local space with a new, dynamically computed covariance $\boldsymbol{\Sigma}_{\text{local}}' = \mathbf{I} + \sigma\mathbf{J}\mathbf{J}^\top$ (Eq. 17).
>
> To our knowledge, this formulation—pre-filtering an intersection-based primitive by analytically deriving an equivalent object-space covariance from a screen-space filter—is a new contribution that has not been proposed before, not even in classical graphics literature on surface splatting, highlighting its novelty and potential impact beyond just 2DGS.
>
> **2. The World-Space Flat Smoothing Kernel: A Necessary and Non-Trivial Formulation for Planar Primitives.**
>
> Our second component addresses pre-aliasing by band-limiting the representation, conceptually inspired by the frequency analysis in Mip-Splatting. However, a 3D volumetric filter cannot be applied to a 2D planar primitive. Our contribution was to correctly formulate this regularization for a flattened representation by projecting the 3D isotropic smoothing kernel onto the 2D splat plane. This results in our effective planar covariance matrix ($\mathbf{V}_k^{\text{eff}}$ in Eq. 9) and corresponding opacity modulation (Eq. 10), which is a necessary step to complete the anti-aliasing framework for 2DGS, especially for the case of magnification.
>
> **3. We provide new insights into the limitations of the original 2DGS.**
>
> A key contribution of our work is the analysis and demonstration of the clamping heuristic's ineffectiveness in the original 2DGS. Prior work had not identified this as a critical source of artifacts. Our experiments (2DGS w/o Clamping in Tables 1-5, and in the Appendix) provides the first concrete evidence that this heuristic often exacerbates aliasing and damages performance, especially during minification. This insight itself is a valuable contribution to the understanding of the vanilla 2DGS method.
>
> ### On the Surface Reconstruction Evaluation
>
> We thank the reviewer for their question regarding surface reconstruction on other datasets. We would like to gently point to **Section 4.5 and Table 4**, where we evaluate our method on the standard DTU surface reconstruction benchmark precisely to address this concern.
>
> The purpose of this experiment was to demonstrate that our anti-aliasing solution preserves the state-of-the-art geometric accuracy that is a major strength of 2DGS. The results in Table 4, showing our method is on par with the original 2DGS confirm that our framework successfully integrates advanced anti-aliasing without degrading this crucial capability. We believe the comprehensive DTU benchmark is sufficient to robustly support this claim.
>
> Our paper introduces a novel, non-trivial method for 2DGS anti-aliasing, provides new insights into the limitations of the original 2DGS, and validates that our approach preserves its key geometric strengths. We hope these clarifications help the reviewer better appreciate the originality and significance of our contributions.
>
> Thank you again for your time and consideration.

---

### Official Review · Reviewer_oTXT · 2025-07-02

**Clarity:** 4
**Significance:** 4
**Originality:** 3
**Rating:** 5
**Confidence:** 3

**Summary:**

The paper extends mip-splatting to 2D Gaussian Splatting (GS) representations. It introduces a world-space flat smoothing kernel that constrains the frequency content of 2D Gaussian primitives based on the maximum sampling frequency of the training views. Additionally, the authors derive an object-space mipmap filter by leveraging an affine approximation of the ray–splat intersection mapping, which enables efficient and accurate anti-aliasing directly in the local space of each splat. Quantitative evaluations show performance comparable to mip-splatting.

**Questions:**

Addressing points 1-2 in weakness is helpful

**Ethical Concerns:**

["NO or VERY MINOR ethics concerns only"]

**Final Justification:**

I have read the rebuttal and the feedback from other reviewers. Shared concerns include the novelty of the approach and the lack of comparison and justification regarding geometric accuracy. I also agree that the formulation appears generally similar to Mip-NeRF’s splatting, but I acknowledge the practical value of adapting Mip-splatting to 2D Gaussian splatting. Addressing aliasing is an important component, especially if the community remains interested in leveraging 2D-GS. Given this, I tend to maintain my positive rating.

**Limitations:**

missing discussion of limitations.

**Quality:**

4

**Strengths And Weaknesses:**

**Strength**
The paper is well-written and addresses an important problem: extending anti-aliasing to 2D Gaussian Splatting representations. This has the potential for significant impact given the widespread use of 2D GS. The derivation is clearly presented, and the results demonstrate state-of-the-art performance for both training and rendering across multiple resolutions.

**Weakness (Minor)**
1. The paper lacks a discussion of its limitations, which would help contextualize the scope and applicability of the proposed method.
2. Including additional visualizations beyond RGB—such as meshes, surface normals, and depth—would strengthen the qualitative analysis. These are particularly relevant since improved geometric modeling is a key motivation for adopting 2D GS.
3. There are broken links in Line 277

---

> ### Author Rebuttal · Authors · 2025-07-30
>
> We sincerely thank the reviewer for their time and for providing a thoughtful and positive assessment of our work. We are grateful for the "excellent" ratings in Quality, Clarity, and Significance and for the recognition that our paper addresses an important problem with a clear derivation and state-of-the-art results.
>
> We would like to address the minor weaknesses pointed out by the reviewer, which we believe will further strengthen our paper.
>
> 1.  **Regarding the Discussion of Limitations:**
>
>     The reviewer noted that "The paper lacks a discussion of its limitations." We would like to respectfully clarify that we included a dedicated **"Limitations" section (Section C) in the Appendix**.
>
>     In this section, we discuss:
>     *   The inherent trade-off between antialiasing and preserving high-frequency detail, a common challenge for all such methods.
>     *   The fundamental limitations stemming from the use of 2D planar primitives, including the "needle-like" artifacts that can appear at grazing angles during magnification.
>     *   The specific trade-offs of our two-part solution, where the Object-Space Mip filter primarily addresses minification (zoom-out) and the Flat Smoothing Kernel targets magnification (zoom-in).
>
>     We acknowledge that this section is in the appendix, which may have been overlooked due to this placement. We believe this discussion thoroughly contextualizes the scope and applicability of our method as requested.
>
> 2.  **Regarding Additional Visualizations (Depth, Normals, Meshes):**
>
>     We thank the reviewer for this constructive suggestion. Our evaluation was centered on showcasing the primary contributions: significant improvements in anti-aliased rendering at multiple scales, while numerically demonstrating the preservation of 2DGS's geometric fidelity on the DTU benchmark. We will happily include additional visualizations in the revision. As the reviewer intuits, the benefits of our anti-aliasing are not confined to RGB output but naturally extend to all rendered buffers.

---

> > ### Comment · Reviewer_oTXT · 2025-08-07
> >
> > The rebuttal addressed most of the concerns. I will maintain my score.

---

### Official Review · Reviewer_MET9 · 2025-07-03

**Clarity:** 3
**Significance:** 3
**Originality:** 2
**Rating:** 4
**Confidence:** 3

**Summary:**

This paper takes insight from Mip-Splatting to bridge the anti aliasing gap between 2dgs and 3dgs. Although 2dgs has better geometry fidelity the clamping method usually leads to the loss of high frequency details. In this paper a smoothing kernel is designed based on the sampling rate of the training views as well as adapting the mip filter to the local space of each splat.

The results show significant improvement upon 2DGS while being compareable agianst mipsplatting in most scenarios.

**Questions:**

How do you compare against hdgs?

**Ethical Concerns:**

["NO or VERY MINOR ethics concerns only"]

**Final Justification:**

I have read the author's reply regarding the specifics of the contributions in this work and other reviews similar concern about the novelty. Given that the steps are crucial for improving 2dgs and results show significant improvement while not adding much overhead I believe the slight lack of novelty is not a justification for rejection of this paper. You don't always need to reinvent the wheel from scratch. Hence maintaining my rating as borderline accept.

**Limitations:**

yes

**Quality:**

3

**Strengths And Weaknesses:**

Strengths
Effective: The core contribution of mitigating aliasing artifacts in 2DGS is highly successful, as evidenced by the significant quantitative improvements against 2DGS across various multi-scale rendering tests (Tables 1, 2, and 3) and compelling qualitative results (Figures 1 and 3). This directly addresses a critical limitation of 2DGS, enhancing its practical applicability.

Computational Efficiency: The paper states that it incurs "a slight overhead of 10-15% in rendering time compared to the original 2DGS," which is a reasonable trade-off for the notable quality improvement. This efficiency ensures the method remains viable for real-time applications.

Preservation of Geometric Fidelity: A key advantage of 2DGS is its superior geometric accuracy. The paper demonstrates that AA-2DGS maintains these capabilities, showing comparable or slightly improved reconstruction performance on the DTU dataset (Table 4), indicating that the anti-aliasing mechanisms do not compromise the underlying geometric representation.

Well-Written and Clear: The paper is well-structured and clearly explains the technical challenges and the proposed solutions. The methodology, particularly the derivation of the object-space Mip filter, is presented in a comprehensible manner.

Weakness:
Incremental Novelty: While highly effective and well-executed, the primary novelty of this work lies in adapting and integrating existing anti-aliasing principles (specifically from mip-splatting and EWA filtering) into the 2Dgs framework. The conceptual methods are largely derived from prior work rather than introducing fundamentally new algorithmic approaches.

nit: line 277: wrong reference ?? , line 278: Such should be lower cased.

---

> ### Author Rebuttal · Authors · 2025-07-30
>
> We thank the reviewer for their positive and constructive feedback. We are pleased that they found our method to be "highly successful", "computationally efficient" and "well-written". We appreciate the opportunity to address the remaining concerns regarding novelty and the comparison to HDGS.
>
> ### On Novelty and Contribution
>
> We respectfully wish to clarify that our work introduces a novel algorithmic framework derived from a signal-processing analysis of the unique 2DGS rendering pipeline and the planar nature of its primitives. This required deriving a first-of-its-kind Object-Space Mip Filter and a non-trivial adaptation of band-limiting for planar primitives with the World-Space Flat Smoothing Kernel, as direct adaptation of prior art is not possible.
>
>
> **1. A Novel Framework is Required for the 2DGS Rendering Pipeline.**
> The rendering mechanisms of 3DGS and 2DGS are fundamentally incompatible, precluding any direct application of 3DGS-based solutions.
> *   **3DGS** projects 3D Gaussians to screen space, which is a well-defined affine transformation. Applying screen-space EWA filtering is a direct extension of existing literature.
> *   **2DGS** does not project primitives. Instead, it computes a non-linear ray-splat intersection (Eq. 5 and 6) to find a point in the planar splat’s local 2D space.
>
> It is not at all trivial how to perform frequency-based band-limiting or pre-filtering in this context. Our "Object-Space Mip Filter" is a novel solution tailored specifically for this unique intersection-based rendering paradigm. **It is a new mathematical derivation with broader impact.**
>
> The reviewer notes that our work adapts existing principles. However, our technical approach is fundamentally different and more novel than that of Mip-Splatting. Mip-Splatting uses the standard screen-space EWA formulation and simply modifies the filter kernel size.
>
> In contrast, our work introduces a new mathematical derivation for an anti-aliasing filter that operates directly in the object space of each splat.
>
> The derivation proceeds as follows:
> *   **The Mapping:** We begin with the ray-splat intersection function $m(\mathbf{x}) = \mathbf{u}(\mathbf{x})$ on the splat (Eq. 5 and 6).
> *   **The Core Innovation:** We form a first-order Taylor approximation of this complex mapping around a pixel, defined by its Jacobian $\mathbf{J} = \frac{\partial \mathbf{u}}{\partial \mathbf{x}}$ (Eq. 12). This local affine approximation is the key insight that makes an analytical solution possible.
> *   **Mapping to Screen Space:** Using this affine mapping and a fundamental property of Gaussian functions, we can express how the local unit Gaussian behaves in screen space (Eq. 13).
> *   **Filtering in Screen Space:** We then mathematically convolve this transformed Gaussian with a screen-space Mip filter (a Gaussian with covariance $\sigma\mathbf{I}$) to obtain a filtered Gaussian with covariance $\mathbf{J}^{-1}\mathbf{J}^{-\top} + \sigma\mathbf{I}$ (Eq. 15).
> *   **Mapping Back to Object Space:** Finally, we use the same affine transform properties to map this filtered screen-space Gaussian back into the splat's local `uv`-space. This yields our final, novel result: the primitive can be evaluated as a single Gaussian in its local space with a new, dynamically computed covariance $\boldsymbol{\Sigma}_{\text{local}}' = \mathbf{I} + \sigma\mathbf{J}\mathbf{J}^\top$ (Eq. 17).
>
> To our knowledge, this formulation—pre-filtering an intersection-based primitive by analytically deriving an equivalent object-space covariance from a screen-space filter—is a new contribution that has not been proposed before, not even in classical graphics literature on surface splatting, highlighting its novelty and potential impact beyond just 2DGS.
>
> **2. The World-Space Flat Smoothing Kernel: A Necessary and Non-Trivial Formulation for Planar Primitives.**
>
> Our second component addresses pre-aliasing by band-limiting the representation, conceptually inspired by the frequency analysis in Mip-Splatting. However, a 3D volumetric filter cannot be applied to a 2D planar primitive. Our contribution was to correctly formulate this regularization for a flattened representation by projecting the 3D isotropic smoothing kernel onto the 2D splat plane. This results in our effective planar covariance matrix ($\mathbf{V}_k^{\text{eff}}$ in Eq. 9) and corresponding opacity modulation (Eq. 10), which is a necessary step to complete the anti-aliasing framework for 2DGS, especially for the case of magnification.
>
> **3. We provide new insights into the limitations of the original 2DGS.**
>
> A key contribution of our work is the analysis and demonstration of the clamping heuristic's ineffectiveness in the original 2DGS. Prior work had not identified this as a critical source of artifacts. Our experiments (2DGS w/o Clamping in Tables 1-5, and in the Appendix) provides the first concrete evidence that this heuristic often exacerbates aliasing and damages performance, especially during minification. This insight itself is a valuable contribution to the understanding of the vanilla 2DGS method.
>
> ### Regarding the Comparison to HDGS
>
> We thank the reviewer for bringing HDGS to our attention. We are happy to provide a direct comparison based on the results presented in both papers.
>
> The HDGS paper includes a "Reduced resolution rendering on the NeRF synthetic dataset" experiment (Table 1 in HDGS), which directly corresponds to our "Single-scale Training and Multi-scale Testing on the Blender Dataset" experiment (Table 2 in our submission). A direct comparison of the average PSNR demonstrates the superiority of our approach:
> *   **Our Method (AA-2DGS):** 32.11 PSNR
> *   **HDGS:** 29.45 PSNR
>
> Our method significantly outperforms HDGS on this benchmark by over 2.6 dB. We believe this substantial performance gap stems from the fundamental difference in our anti-aliasing strategies. HDGS employs a frustum-based super-sampling method. It treats each pixel as a frustum and casts 5 rays to average the result which is approximative and computationally more expensive. Differently, our method uses a principled analytical pre-filtering approach that mitigates aliasing more effectively and efficiently by directly addressing its signal-processing roots.
>
> Finally, we will gladly correct any typos and formatting issues in the final version.
>
> We hope these clarifications have addressed the reviewer's concerns and reinforced the novelty and significance of our contributions. We thank the reviewer again for their valuable time and feedback.

---

### Decision · Program_Chairs · 2025-09-17

**Decision:**

Accept (poster)

**Comment:**

The final recommendations of the expert reviewers are 2x Borderline accept, 1x Accept, and 1x Borderline reject. These recommendations are made after addressing the authors' feedback and through a thorough discussion. This paper presents a framework to bridge the antialiasing gap between 2D Gaussian Splatting (2DGS) and 3D Gaussian Splatting (3DGS), drawing key inspiration from Mip-Splatting. The core contribution lies in the derivation of an object-space mipmap filter through an affine approximation of the ray-splat intersection mapping. This enables efficient and accurate antialiasing directly within the local coordinate space of each splat, offering a principled approach to addressing artifacts that are particularly prominent in 2DGS. The technical approach is well-motivated, addressing a known and important problem in 3DGS. The authors present compelling visual results and quantitative comparisons that demonstrate improved image quality compared to the baseline 2DGS.

However, all the reviewers raise concerns regarding the perceived novelty and technical depth of the contribution. The reviewers are not fully convinced that the work is more than a straightforward application of the principles from Mip-Splatting to the 2DGS framework. The “why” and “how” of the proposed object-space affine approximation, and what makes it a non-trivial step, are not sufficiently explained. The reviewers have also pointed out that the geometric accuracy is insufficient, which is a known strength of 2DGS. In addition, the discussion of limitations is misplaced within the paper’s structure and should be moved to a more standard position (e.g., before the conclusion).

The Area Chair concurs with the reviewers that the concerns raised are valid and must be addressed for the paper to realize its full potential. Despite these concerns, the Area Chair believes the paper makes a valuable contribution to the field of Gaussian-Splatting-based techniques. Given the overall strength of the method and the thoughtful response to reviewer feedback, the consensus supports acceptance.